# Nitrogen Fixation in Arctic Coastal Waters (Qeqertarsuaq, West Greenland): Influence of Glacial Melt on Diazotrophs, Nutrient Availability, and Seasonal Blooms

Schlangen Isabell[1], Leon-Palmero Elizabeth[1,2], Moser Annabell[1], Xu Peihang[1], Laursen Erik[1], and Löscher Carolin R.[1,3]

[1]Nordcee, Department of Biology, University of Southern Denmark, Campusvej 55, 5230 Odense M, Denmark
[2]Department of Geosciences, Princeton University, Princeton, New Jersey
[3]DIAS, University of Southern Denmark, Odense, Denmark

**Correspondence:** Carolin R. Löscher (cloescher@biology.sdu.dk)

**Abstract.** The Arctic Ocean is undergoing rapid transformation due to climate change, with decreasing sea ice contributing to a predicted increase in primary productivity. A critical factor determining future productivity in this region is the availability of nitrogen, a key nutrient that often limits biological growth in Arctic waters. The fixation of dinitrogen ($N_2$) gas, a biological process mediated by diazotrophs, provides a source of new nitrogen to marine ecosystems and has been increasingly recognized as a potential contributor to nitrogen supply in the Arctic Ocean. Historically it was believed to be limited to oligotrophic tropical and subtropical oceans, Arctic $N_2$ fixation has only garnered significant attention over the past decade, leaving a gap in our understanding of its magnitude, the diazotrophic community, and potential environmental drivers. In this study, we investigated $N_2$ fixation rates and the diazotrophic community in Arctic coastal waters, using a combination of isotope labeling, genetic analyses and biogeochemical profiling, in order to explore its response to glacial meltwater, nutrient availability and its impact on primary productivity. We observed $N_2$ fixation rates ranging from 0.16 to 2.71 nmol N $L^{-1}$ $d^{-1}$, notably higher than many previously reported rates for Arctic waters. The diazotrophic community was predominantly composed of UCYN-A. The highest $N_2$ fixation rates co-occurred with peaks in chlorophyll *a* and primary production at a station in the Vaigat Strait, likely influenced by glacial meltwater input. On average, $N_2$ fixation contributed 1.6% of the estimated nitrogen requirement of primary production, indicating that while its role is modest, it may still represent a nitrogen source in certain conditions. These findings illustrate the potential importance of $N_2$ fixation in an environment previously not considered important for this process and provide insights into its response to the projected melting of the polar ice cover.

## 1 Introduction

Nitrogen is a key element for life and often acts as a growth-limiting factor for primary productivity (Gruber and Sarmiento, 1997; Gruber, 2004; Gruber and Galloway, 2008). Despite nitrogen gas ($N_2$) making up approximately 78% of the atmosphere, it remains inaccessible to most marine life forms. Diazotrophs, which are specialized bacteria and archaea, have the ability to convert $N_2$ into biologically available nitrogen, facilitated by the nitrogenase enzyme complex carrying

out the process of
biological nitrogen fixation ($N_2$ fixation) (Capone and Carpenter (1982)). Despite the fact that these organisms are highly
spe- cialized and $N_2$ fixation is energetically demanding, the ability to carry out this process is widespread amongst
prokaryotes. However, it is controlled by several factors such as temperature, light, nutrients and trace metals such as iron
and molybdenum (Sohm et al., 2011; Tang et al., 2019). Oceanic $N_2$ fixation is the major source of nitrogen to the marine
system (Karl et al., 2002; Gruber and Sarmiento, 1997), thus, diazotrophs determine the biological productivity of our
planet (Falkowski et al. (2008), impact the global carbon cycle and the formation of organic matter (Galloway et al., 2004;
Zehr and Capone, 2020). Traditionally it has been believed that the distribution of diazotrophs was limited to warm and
oligotrophic waters (Buchanan et al., 2019; Sohm et al., 2011; Luo et al., 2012) until putative diazotrophs were identified
in the central Arctic Ocean and Baffin Bay (Farnelid et al., 2011; Damm et al., 2010). First rate measurements have been
reported for the Canadian Arctic by Blais et al. (2012) and recent studies have reported rate measurements in adjacent seas
(Harding et al., 2018; Sipler et al., 2017; Shiozaki et al., 2017, 2018), drawing attention to cold and temperate waters as
significant contributors to the global nitrogen budget through diverse organisms.
UCYN-A has been described as the dominant active $N_2$ fixing cyanobacterial diazotroph in arctic waters (Harding et al.
(2018)), while other cyanobacteria have only occasionally been reported (Díez et al., 2012; Fernández-Méndez et al., 2016;
Blais et al.,). However, other recent studies suggest, that the majority of the arctic marine diazotrophs are NCDs (non-
cyanobacterial diazotroph) and those may contribute significantly to $N_2$ fixation in the Arctic Ocean (Shiozaki et al., 2018;
Fernández-Méndez et al., 2016; Harding et al., 2018; Von Friesen and Rie-mann, 2020). Recent work by Robicheau et al.
(2023) nearby Baffin Bay, geographically close to the sampling area, document low *nif*H gene abundance while still detecting
diazotrophs in Arctic surface waters, highlighting the patchy distribution of diazotrophs across Arctic coastal environments. Studies
on the Arctic diazotroph community remain scarce, leaving Arctic environments poorly understood regarding $N_2$ fixation.
Shao et al. (2023) note the impossibility of estimating Arctic $N_2$ fixation rates due to the sparse spatial coverage, which
currently represents only approximately 1 % of the Arctic Ocean. Increasing data coverage in future studies will aid in
better constraining the contribution of $N_2$ fixation to the global oceanic nitrogen budget (Tang et al. (2019)).
The Arctic ecosystem is undergoing significant changes driven by rising temperatures and the accelerated melting of sea ice,
a trend predicted to intensify in the future (Arrigo et al., 2008; Hanna et al., 2008; Haine et al., 2015). These climate-driven
shifts have stimulated primary productivity in the Arctic by 57 % from 1998 to 2018,  elevating nutrient demands in the
Arctic Ocean (Ardyna and Arrigo, 2020; Arrigo and van Dijken, 2015; Lewis et al., 2020). This increase is attributed to
prolonged phytoplankton growing seasons and expanding ice-free areas suitable for phytoplankton growth (Arrigo et al.
(2008)). However, despite these dramatic changes, the role of $N_2$ fixation in sustaining Arctic primary production remains
poorly understood. While recent studies suggest that diazotrophic activity may contribute to nitrogen inputs in polar regions
(Sipler et al. (2017)), fundamental uncertainties remain regarding the extend, distribution and environmental drivers of $N_2$
Fixation in the Arctic Ocean. Specifically, it is unclear whether increased glacial meltwater input enhances or inhibits $N_2$
Fixation through changes in nutrient availability, stratification, and microbial community composition. Thus, the question
of whether nitrogen limitation will emerge as a key factor constraining Arctic primary production under future climate scenarios
remains unresolved. In this study, we investigate the diversity of diazotrophic communities alongside in situ $N_2$ fixation
rate measurements in Disko Bay (Qeqertarsuaq), a coastal Arctic system strongly influenced by glacial meltwater input. By linking
environmental parameters to $N_2$ fixation dynamics, we aim to clarify the role of diazotrophs in Arctic nutrient cycling and
assess their potential contribution to sustaining primary production in a changing Arctic. Understanding these processes is
essential for refining biogeochemical models and predicting ecosystem responses to future climate change.
## 2   Material and methods
### 2.1   Seawater sampling
The research expedition was conducted from August 16 to 26 in 2022 aboard the Danish military vessel P540 within the
waters of Qeqertarsuaq, located in the western region of Greenland (Kalaallit Nunaat). Discrete water samples were
obtained using a 10 L Niskin bottle, manually lowered with a hand winch to five distinct depths (surface, 5, 25, 50, and
100 m). A comprehensive sampling strategy was employed at 10 stations (Fig. 1), covering the surface to a depth of 100 m.
The sampled parameters included water characteristics, such as nutrient concentrations, chl $a$, particulate organic carbon
(POC) and nitrogen (PON), molecular samples for nucleic acid extractions (DNA), dissolved inorganic carbon (DIC) as
well as CTD sensor data. At three selected stations (3,7,10) $N_2$ fixation and primary production rates were quantified
through concurrent incubation experiments.
Samples for nutrient analysis, nitrate ($NO_3^-$), nitrite ($NO_2^-$) and phosphate ($PO_4^{3-}$) were taken in triplicates, filtered
through a 0.22 $\mu$m syringe filter (Avantor VWR® Radnor, Pa, USA) and stored at -20 °C until further analysis.
Concentrations were spectrophotometrically determined (Thermo Scientific, Genesys 1OS UV-VIS spectrophotometer)
following the established protocols of Murphy and Riley (1962) for $PO_4^{3-}$; García-Robledo et al. (2014) for $NO_3^-$ & $NO_2^-$
( detection limits: 0.01 μmol L$^{-1}$ ($NO_3^-$, $NO_2^-$, and $PO_4^{3-}$), 0.05 μmol L$^{-1}$ ($NH_4^+$). Chl $a$ samples were filtered onto 47 mm
ø GF/F filters (GE Healthcare Life Sciences, Whatman, USA), placed into darkened 15 mL LightSafe centrifuge tubes
(Merck, Rahway, NJ, USA) and were subsequently stored at -20 °C until further analysis. To determine the Chl $a$
concentration, the samples were immersed in 8 mL of 90 % acetone overnight at 5 °C. Subsequently, 1 mL of the resulting
solution was transferred to a 1.5 mL glass vial (Mikrolab Aarhus A/S, Aarhus, Denmark) the following day and subjected
to analysis using the Triology® Fluorometer (Model #7200-00) equipped with a Chl $a$ in vivo blue module (Model #7200-
043, both Turner Designs, San Jose, CA, USA). Measurements of serial dilutions from a 4 mg L$^{-1}$ stock standard and 90 %
acetone (serving as blank) were performed to calibrate the instrument. In addition, measurements of a solid-state secondary
standard were performed every 10 samples. Water (1 L) from each depth was filtered for the determination of POC and
PON concentrations, as well as natural isotope abundance ($\delta$ $^{13}$C POC / $\delta$ $^{15}$N PON) using 47 mm ø, 0.7 $\mu$m nominal pore
size precombusted GF/F filter (GE Healthcare Life Sciences, Whatman, USA), which were subsequently stored at -20 °C
until further analysis. Seawater samples for DNA were filtered through 47 mm ø, 0.22 $\mu$m MCE membrane filter (Merck,
Millipore Ltd., Ireland) for a maximum of 20 minutes, employing a gentle vacuum (200 mbar). The filtered volumes varied
depending
on the amount of material captured on the filter, ranging from 1.3 L to 2 L, with precise measurements recorded. The filters
were promptly stored at -20 °C on the ship and moved to -80 °C upon arrival to the lab until further analysis.
To achieve detailed vertical profiles, a conductivity-temperature-depth-profiler (CTD, Seabird X) equipped with
supplemen- tary sensors for dissolved oxygen (DO), photosynthetic active radiation (PAR), and fluorescence (Flourometer)
was manually deployed.

## 111 2.2 Nitrogen fixation and primary production


Water samples were collected at three distinct depths (0, 25 and 50 m) for the investigation of $N_2$ fixation rates and primary
production rates, encompassing the euphotic zone, chlorophyll maximum, and a light-absent zone. Three incubation
stations (Fig. 2: station 3, 7, 10) were chosen, in a way to cover the variability of the study area. This strategic sampling
aimed to capture a gradient of the water column with varying environmental conditions, relevant to the aim of the study.
$N_2$ fixation rates were assessed through triplicate incubations employing the modified $^{15}N$-$N_2$ dissolution technique after
Großkopf et al. (2012) and Mohr et al. (2010).
To ensure minimal contamination, 2.3 L glass bottles (Schott-Duran, Wertheim, Germany) underwent pre-cleaning and
acid washing before being filled with seawater samples. Oxygen contamination during sample collection was mitigated by
gently and bubble-free filling the bottles from the bottom, allowing the water to overflow. Each incubation bottle received
a 100 mL amendment of $^{15}N$-$N_2$ enriched seawater (98 %, Cambridge Isotope Laboratories, Inc.,USA) achieving an average
dissolved $N_2$ isotope abundance ($^{15}N$ atom %) of 3.90 ± 0.02 atom % (mean ± SD). Additionally, 1 mL of $H^{13}CO_3$ (1g/50
mL) (Sigma- Aldrich, Saint Louis Missouri US) was added to each incubation bottle, roughly corresponding to 10 atom %
enrichment and thus measurements of primary production and $N_2$ fixation were conducted in the same bottle. Following
the addition of both isotopic components, the bottles were closed airtight with septa-fitted caps and incubated for 24 hours
on-deck incubators with a continuous surface seawater flow. These incubators, partially shaded (using daylight-filtering
foil) to simulate in situ photosynthetically active radiation (PAR) conditions, aimed to replicate environmental parameters
experienced at the sampled depths. Control incubations utilizing atmospheric air served as controls to monitor any natural
changes in $\delta$ $^{15}N$ not attributable to $^{15}N$-$N_2$ addition. These control incubations were conducted using the dissolution
method, like the $^{15}N$-$N_2$ enrichment experiments, but with the substitution of atmospheric air instead of isotopic tracer.
After the incubation period, subsamples for nutrient analysis were taken from each incubation sample, and the remaining
content was subjected to the filtration process and were gently filtered (200 mbar) onto precombusted GF/F filters
(Advantec,
47 mm ø, 0.7 $\mu$m nominal pore size). This step ensured a comprehensive examination of both nutrient dynamics and the
isotopic composition of the particulate pool in the incubated samples. Samples were stored at -20 °C until further analysis.
Upon arrival in the lab, the filters were dried at 60 °C and to eliminate particulate inorganic carbon, subsequently subject to
acid fuming during which they were exposed to concentrated hydrochloric acid (HCL) vapors overnight in a desiccator. After
undergoing acid treatment, the filters were carefully dried, then placed into tin capsules and pelletized for subsequent analysis.
The determination of POC and PON, as well as isotopic composition ($\delta$ $^{13}$C POC / $\delta$ $^{15}$N PON), was carried out using an
elemental analyzer (Flash EA, ThermoFisher, USA) connected to a mass spectrometer (Delta V Advantage Isotope Ratio MS,
ThermoFisher, USA) with the ConFlo IV interface. This analytical setup was applied to all filters. These values, derived from
triplicate incubation measurements, exhibited no omission of data points or identification of outliers. Final rate calculations for
$N_2$ fixation rates were performed after Mohr et al. (2010) and primary production rates after Slawyk et al. (1977). A detailed
sensitivity analysis of $N_2$ fixation rates, including the contribution of each source of error for all parameters, is provided in a
supplementary table and summarized form in the Appendix (Table A1).
**2.3   Molecular methods**
The filters were flash-frozen in liquid nitrogen, crushed and DNA was extracted using the Qiagen DNA/RNA AllPrep Kit
(Qi- agen, Hildesheim, DE), following the procedure outlined by the manufacturer. The concentration and quality of the
extracted DNA was assessed spectrophotometrically using a MySpec spectrofluorometer (VWR, Darmstadt, Germany).
The prepara- tion of the metagenome library and sequencing were performed by BGI (China). Sequencing libraries were
generated using MGIEasy Fast FS DNA Library Prep Set following the manufacturer's protocol. Sequencing was
conducted with 2x150bp on a DNBSEQ-G400 platform (MGI). SOAPnuke1.5.5 (Chen et al. (2018)) was used to filter and
trim low quality reads and adaptor contaminants from the raw sequence reads, as clean reads. In total, fifteen metagenomic
datasets were produced with an average of 9.6G bp per sample.
**2.3.1   Metagenomic De Novo assembly, gene prediction, and annotation**
Megahit v1.2.9 (Li et al. (2015)) was used to assemble clean reads for each dataset with its minimum contig length as 500.
Prodigal v2.6.3 (Hyatt et al. (2010)) with the setting of "-p meta" was then used to predict the open reading frames (ORFs)
of the assembled contigs. ORFs from all the available datasets were filtered (>100bp), dereplicated and merged into a
catalog of non-redundant genes using cd-hit-est (>95 % sequence identity) (Fu et al. (2012)). Salmon v1.10.0 (Patro et al.
(2017)) with the "– meta" option was employed to map clean reads of each dataset to the catalog of non-redundant genes
and generate the GPM (genes per million reads) abundance. Eggnog mapper v2.1.12 (Cantalapiedra et al. (2021)) was then
performed to assign KEGG Orthology (KO) and identify specific functional annotation for the catalog of non-redundant
genes. The marker genes, *nifD*K (K02586, K02591 nitrogenase molybdenum-iron protein alpha/beta chain) and *nif* H
(K02588, nitrogenase iron protein), were used for the evaluation of microbial potential of $N_2$ fixation. *Rbc*L (K01601,
ribulose-bisphosphate carboxylase large chain) and *psb*A (K02703, photosystem II P680 reaction center D1 protein) were
selected to evaluate the microbial potential of carbon fixation and photosynthesis, respectively. The molecular datasets
have been deposited with the accession number: Bioproject PRJNA1133027.
**3   Results and discussion**
**3.1 Hydrographic conditions in Qeqertarsuaq (Disco Bay) and Sullorsuaq (Vaigat) Strait**
Disko Bay (Qeqertarsuaq) is located along the west coast of Greenland (Kalaallit Nunaat) at approximately 69 °N (Figure
1), and is strongly influenced by the West Greenland Current (WGC) which is associated with the broader Baffin Bay Polar
Waters (BBPW) (Mortensen et al., 2022; Hansen et al., 2012). The WGC does not only significantly shape the hydrographic
conditions within the bay but also plays an important role in the larger context of Greenland Ice Sheet melting (Mortensen
et al. (2022)). Central to the hydrographic system of the Qeqertarsuaq area is the Jakobshavn Isbræ, which is the most
productive glacier in the northern hemisphere and believed to drain about 7 % of the Greenland Ice Sheet and thus
contributes substantially to the water influx into the Qeqertarsuaq (Holland et al. (2008)). A predicted increased inflow of
warm subsurface water, originating from North Atlantic waters, has been suggested to further affect the melting of the
Jakobshavn Isbræ and thus adds another layer of complexity to this dynamic system (Holland et al., 2008; Hansen et al.,
184 2012).
The hydrographic conditions in Qeqertarsuaq have a significant influence on biological processes, nutrient availability, and the

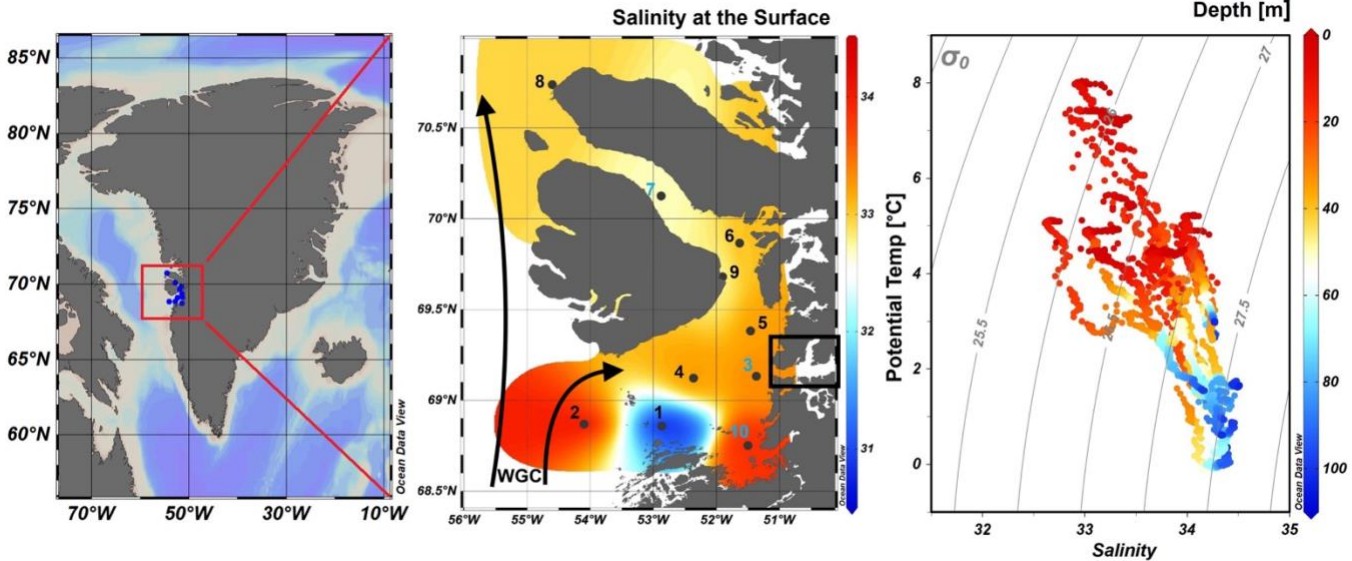


**Figure 1.** Map of Greenland (Kalaallit Nunaat) with indication of study area (red box), on the left. Interpolated distribution of Sea
Surface Salinity (SSS) values with corresponding isosurface lines and indication of 10 sampled stations (normal stations in black,
incubation stations in blue), black arrows indicate the West Greenland Current (WGC) and the black box indicate the location of the
Jakobshavn Isbræ, in the middle. Scatterplot of the potential temperature and salinity for all station data. The plot is used for the
identification of the main water masses within the study area. Isopycnals (kg m⁻³) are depicted in grey lines, on the right. Figures were
created in Ocean Data View (ODV) (Schlitzer (2022)).
broader marine ecosystem (Munk et al., 2015; Hendry et al., 2019; Schiøtt, 2023).
During our survey, we found very heterogenous hydrographic conditions at the different stations across Qeqertarsuaq (Fig. 1
& Fig. 2). The three selected stations for $N_2$ fixation analysis (stations 3, 7, and 10) were strategically chosen to capture the
spatial
variability of the area. Surface salinity and temperature measurements at these stations indicate the influence of freshwater
input. The surface temperature exhibit a range of 4.5 to 8 °C, while surface salinity varies between 31 and 34, as illustrated
in Fig. 1. The profiles sampled during our survey extend to a maximum depth of 100 m. Comparison of temperature/salinity
(T/S) plots with recent studies suggests the presence of previously described water masses, including Warm Fjord Water
(WFjW) and Cold Fjord Water (CFjW) with an overlaying surface glacial meltwater runoff. Those water masses are defined
with a density range of $27.20 \leq \sigma_\theta \leq 27.31$ but different temperature profiles. Thus water masses can be differentiated by
their temperature within the same density range (Gladish et al. (2015)). Other water masses like upper subpolar mode water
(uSPMW), deep subpolar mode water (dSPMW) and Baffin Bay polar Water (BBPW) which has been identified in the
Disko Bay (Qeqertarsuaq) before, cannot be identified from this data and may be present in deeper layers (Mortensen et
al., 2022; Sherwood et al., 2021; Myers and Ribergaard, 2013; Rysgaard et al., 2020). The temperature and salinity profiles
across the 10

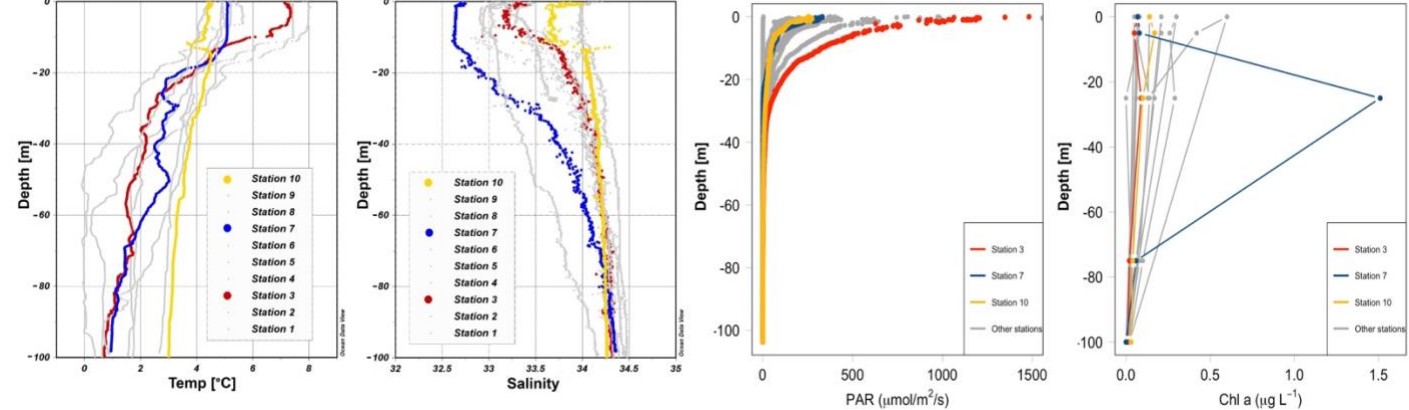

**Figure 2.** Profiles of temperature (°C), salinity, photosynthetically active radiation (PAR) (μmol/m²/s) and Chl *a* (mg m⁻³) across stations
1 to 10 with depth (m). Stations 3, 7, and 10 are highlighted in red, blue, and yellow, respectively, to emphasize incubation stations.
Figures were created in Ocean Data View and R-Studio (Schlitzer (2022)).
stations in the study area show distinct stratification and variability, which is represented through the three incubation
stations (highlighted stations 3, 7, and 10 in Fig. 2). They display varying degrees of stratification and mixing, with notable
differences in the salinity and termperature profiles. Station 3 and station 7 exhibit clear stratification in both temperature
and salinity marked by the presence of thermoclines and haloclines. These features suggest significant freshwater input
influenced by local weather conditions and climate dynamics, like surface heat absorption. In contrast, Station 10 exhibits a
narrower range of temperature and salinity values throughout the water column compared to Stations 3 and 7, indicating
more well-mixed conditions. This uniformity is likely influenced by the regional circulation pattern and partial upwelling
(Hansen et al., 2012; Krawczyk et al., 2022). The distinct characteristics observed at station 10, as illustrated in the surface
plot (Fig. 1), show an elevated salinity and colder temperatures compared
to the other stations. This feature suggests upwelling of deeper waters along the shallower shelf, likely facilitated by the
local seafloor topography. Specifically, the seafloor shallowing off the coast of Station 10 may act as a barrier, disrupting
typical circulation and forcing deeper, saltier, and colder waters to the surface. This pattern aligns with previous studies that
describe similar mechanisms in the region (Krawczyk et al. (2022)). Their description of the bathymetry in Qeqertarsuaq,
featuring depths ranging from ca. 50 to 900 m, suggests its impact on turbulent circulation patterns, leading to the mixing
of different water masses. Evident variability in oceanographic conditions that can be observed throughout the study area
occurs particularly along characteristic topographical features like steep slopes, canyons, and shallower areas. The summer
melting of sea ice and glaciers introduces freshwater influxes that create distinct vertical and horizontal gradients in salinity
and temperature in the Qeqertarsuaq area Hansen et al. (2012). Additionally, the accelerated melting of the Jakobshavn
Isbræ, influenced by the warmer inflow from the West Greenland Intermediate Current (WGIC), further alters the
hydrographic conditions. Recent observations indicate significant warming and shoaling of the WGIC, potentially enabling
it to overcome the sill separating the Illulissat Fjord from the Qeqertarsuaq area (Hansen et al., 2012; Holland et al., 2008;
Myers and Ribergaard, 2013). This shift intensifies glacier melting, driving substantial changes in the local ecological
dynamics (Ardyna et al., 2014; Arrigo et al., 2008; Bhatia et al., 2013).

### 3.2    $N_2$ Fixation Rate Variability and Associated Environmental Conditions

We quantified $N_2$ fixation rates within the waters of Qeqertarsuaq, spanning from the surface to a depth of 50 m (Table 1).
The rates ranged from 0.16 to 2.71 nmol N $L^{-1}$ $d^{-1}$ with all rates surpassing the minimum quantifiably rate (Appendix
Table 1). Our findings represent rates at the upper range of those observed in the Arctic Ocean. Previous measurements in
the region have been limited, with only one study in Baffin Bay by Blais et al. (2012), reporting rates of 0.02 nmol N $L^{-1}$ $d^{-1}$
$^{1}$, which are 1-2 orders of magnitude lower than our observations. Moreover, Sipler et al. (2017), reported rated in the coastal
Chukchi Sea, with average values of 7.7 nmol N $L^{-1}$ $d^{-1}$. These values currently represent some of the highest rates measured
in Arctic shelf environments. Compared to these, our highest measured rate (2.71 nmol N $L^{-1}$ $d^{-1}$) is lower, but still
important, particularly considering the more Atlantic-influenced location of our study site. Sipler et al. (2017) also noted
that a significant fraction of diazotrophs were <3 μm in size, suggesting that small unicellular diazotrophs play a dominant
role in Arctic nitrogen fixation. Altogether, our data contribute to the growing evidence that $N_2$ fixation is a widespread
and potentially significant nitrogen source across various Arctic regions. Simultaneous primary production rate
measurements ranged from 0.07 to 3.79 $\mu$mol N $L^{-1}$ $d^{-1}$, with the highest rates observed at station 7 and generally higher
values in the surface layers. Employing Redfield stoichiometry, the measured $N_2$ fixation rates accounted for 0.47 to 2.6 %
(averaging 1.57 %) of primary production at our stations. The modest contribution to primary production suggests that $N_2$
fixation does not exert a substantial influence on the productivity of these waters during the time of the sampling. Rather,
our $N_2$ fixation rates suggest primary production to depend mostly on additional nitrogen sources including regenerated,
meltwater or land-based sources.
While the N:P ratio is commonly used to assess nutrient limitations relative to Redfield stoichiometry, most DIN and DIP
measurements in our study were below detection limit (BDL), preventing a reliable calculation for this ratio. As such, we
refrain from drawing conclusions based on N:P stoichiometry. Nevertheless, previous studies by Jensen et al. (1999) and
Tremblay and Gagnon (2009), have identified nitrogen limitation in this region. Such biogeochemical conditions, when
present, would be expected to generate a niche for $N_2$ fixing organisms (Sohm et al. (2011)). While $N_2$ fixation did not
chiefly sustain primary production during our sampling campaign, we hypothesize that $N_2$ fixation has the potential to play
a role in bloom dynamics under certain conditions. As nitrogen availability decreases
during a bloom, it may provide a niche for $N_2$ fixation, potentially helping  to extend the productive period of the bloom
period (Reeder et al. (2021)). Satellite data indicates that a fall bloom began in early August, following the annual spring
bloom, as described by Ardyna et al. (2014). This double bloom situation may be driven by increased melting and the
subsequent input of bioavailable nutrients and iron (Fe) from meltwater runoff (Arrigo et al., 2017; Hopwood et al., 2016;
Bhatia et al., 2013). The meltwater from the Greenland Ice Sheet is a significant source of Fe (Bhatia et al., 2013; Hawkings
et al., 2015, 2014), which is a limiting factor especially for diazotrophs (Sohm et al. (2011)). Consequently, it is plausible
that Fe and nutrients from the Isbræ glacier create favorable conditions for both bloom development and diazotroph activity
in Qeqertarsuaq. However, we emphasize that confirming a causal link between $N_2$ fixation and secondary bloom
development requires further evidence, such as time-series data on nutrient concentrations, diazotroph abundance, and
bloom dynamics.

**Table 1.** $N_2$ fixation (nmol N $L^{-1}$ $d^{-1}$), standard deviation (SD), primary productivity (PP; $\mu$mol C $L^{-1}$ $d^{-1}$), SD, percentage of estimated
new primary productivity (% New PP) sustained by $N_2$ fixation, dissolved inorganic nitrogen compounds ($NO_x$), phosphorus ($PO_4$), and
the molar nitrogen-to-phosphorus ratio (N:P) at stations 3, 7, and 10. BDL= Below detection limit.

| Station (no.) | Depth (m) | $N_2$ fixation (nmol N $L^{-1}$ $d^{-1}$) | SD ($\pm$) | Primary Productivity ($\mu$mol C $L^{-1}$ $d^{-1}$) | SD ($\pm$) | % New PP (%) | $NO_x$ ($\mu$mol $L^{-1}$ $d^{-1}$) | $PO_4$ ($\mu$mol $L^{-1}$ $d^{-1}$) |
|---|---|---|---|---|---|---|---|---|
| 3 | 0 | 1.20 | 0.21 | 0.466 | 0.08 | 1.71 | BDL | BDL |
| 3 | 25 | 1.88 | 0.11 | 0.588 | 0.04 | 2.11 | BDL | 0.70 |
| 3 | 50 | 0.29 | 0.01 | 0.209 | 0.00 | 0.91 | 0.33 | 1.48 |
| 7 | 0 | 2.49 | 0.44 | 0.63 | 0.20 | 2.60 | BDL | BDL |
| 7 | 25 | 2.71 | 0.22 | 3.79 | 2.45 | 0.47 | BDL | 0.45 |
| 7 | 50 | 0.53 | 0.24 | 0.33 | 0.36 | 1.08 | BDL | 0.97 |
| 10 | 0 | 1.48 | 0.12 | 0.74 | 0.15 | 1.33 | BDL | BDL |

| 10 | 25 | 0.31 | 0.01 | 0.29 | 0.07 | 0.73 | BDL | BDL |
| 10 | 50 | 0.16 | 0 | 0.07 | 0.07 | 1.40 | BDL | BDL |

A near-Redfield stoichiometry in POC:PON suggests that the particulate organic matter (POM) likely originates from an
ongoing phytoplankton bloom, as phytoplankton generally assimilate carbon and nitrogen in relatively consistent
proportions during active growth (Redfield 1934). However this assumption is based on a global average, and POM
stoichiometry can exhibit substantial latitudinal variation. Deviations may also arise during particle production and
remineralization processes (Redfield 1934; Geider and La Roche 2002; Sterner and Elser 2017; Quigg et al., 2003). Recent
studies have further shown that POM composition vary widely across plankton communities, influenced by factors such as
growth rates, community composition, ad physiological status (e.g. fast- vs- slow-growing organisms), with degradation
often playing a secondary role (Tanioka et al., 2022). Additionally, terrestrial organic material—likely introduced via glacial
outflow in the study area—may also contribute to the observed POM composition (Schneider et al., 2003). Latitudinal
variability in organic matter stoichiometry has also been linked to differences in nutrient supply and phosphorus stress
(Fagan et al., 2024; Tanioka et al., 2022). Consequently, the near-Redfield stoichiometry observed here cannot be clearly
attributed to freshly produced organic material. Nevertheless, satellite-derived surface chlorophyll *a* concentration and
associated primary production support the interpretation that recently produced organic matter does contribute, at least in
part, to the sinking POM captured in our samples. Since inorganic nitrogen species (e.g., NOx) were below detection limits,
direct calculation or interpretation of the N:P ratio in the dissolved nutrient pool was not possible and has been avoided. The
absence of available nitrogen may nonetheless reflect nitrogen depletion, potentially creating ecological niches for
diazotrophs and nitrogen-fixing organisms. Such conditions may promote shifts in microbial community structure, as
observed by Laso-Perez et al. (2024).   Laso Perez et al. (2024) documented changes in microbial community composition
during an Arctic bloom, focusing on nitrogen cycling. They observed a shift from chemolithotrophic to heterotrophic
organisms throughout the summer bloom and noted increased activity to compete for various nitrogen sources. However, no
*nif* H gene copies, indicative of nitrogen-fixing organisms, were found in their dataset based on metagenome-assembled
genomes (MAGs). This is not unexpected due to the classically low abundance of diazotrophs in marine microbial
communities which has often been described (Turk-Kubo et al., 2015; Farnelid et al., 2019). Given the high productivity
and metabolic activity observed in Qeqertarsuaq during a similar bloom period, the detected diazotrophs (Section 3.3) may
play a more significant role than previously thought. Across the 10 stations there is considerable variability in POC and PON
concentrations (Fig. 3). PON concentrations range from 0.0 µmol N L$^{-1}$ to 3.48 µmol N L$^{-1}$ (n=124), while POC
concentrations range from 2.7 µmol C L$^{-1}$ to 27.2 µmol C L$^{-1}$ (n=144). The highest concentrations for both PON and POC
were observed at station 7 at a depth of 25 m and coincide with the highest reported N$_2$ fixation rate (Figure Appendix A2
& A3). Generally, POC and PON concentrations decrease with depth, peaking at the deep chl *a* maximum (DCM), identified
between 15 to 30 m across all stations. The DCM was identified based on measured chl *a* concentrations and previous
descriptions in the region (Fox and Walker, 2022; Jensen et al., 1999). The variability in chl *a* concentrations indicates

differences in phytoplankton abundance among the stations, with concentrations ranging between 0 to 0.42 mg m$^{-3}$. Excluding station 7, which exhibited the highest chl $a$ concentration at the DCM (1.51 mg m$^{-3}$). While Tang et al. (2019) found that $N_2$ fixation measurements strongly correlated to satellite estimates of chl $a$ concentrations, our results did not show a statistically significant correlation between nitrogen fixation rates and chl a concentrations overall (Figures A2 & A3). However, as noted, Station 7 at 25 m represents a unique case. The elevated concentration of chl a at this station likely resulted from a local phytoplankton bloom induced by meltwater outflow from the Isbræ glacier and sea ice melting, which may help explain the observed nitrogen fixation rates (Arrigo et al., 2017; Wang et al., 2014). This study's findings are in agreement with prior reports of analogous blooms occurring in the region (Fox and Walker, 2022; Jensen et al., 1999).

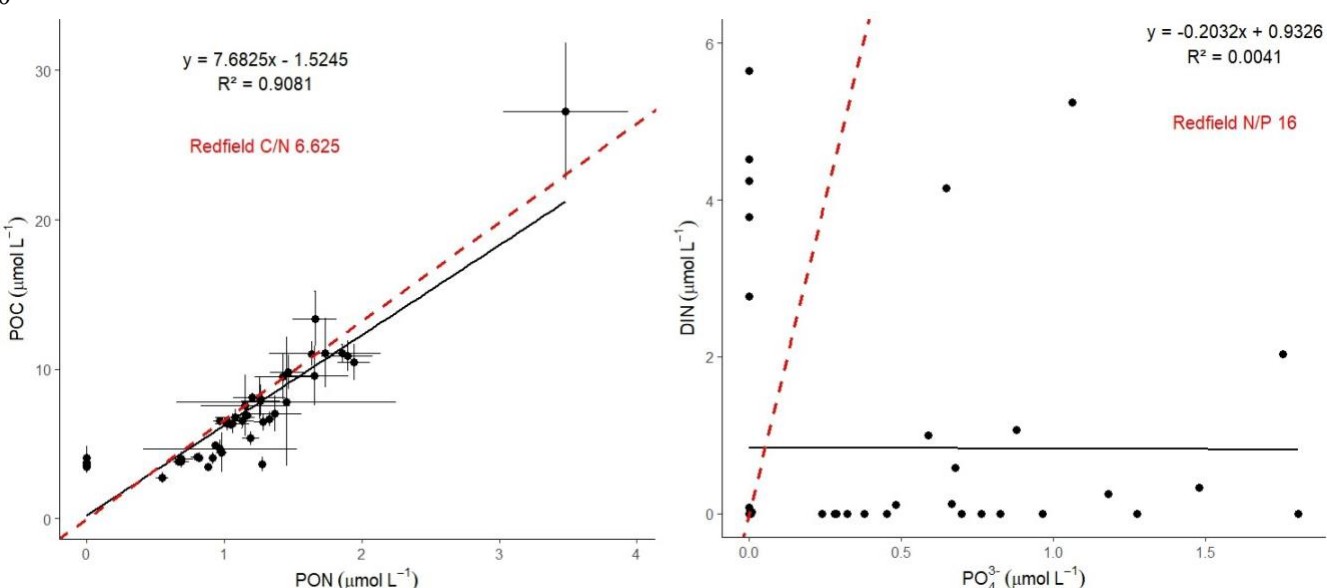

**Figure 3.** The POC/PON and DIN/DIP ratios at all 10 stations. The red line represents the Redfield ratios of POC/PON (106:16) and DIN/DIP (16:1).

### 3.3 Potential Contribution of UCYN-A to Nitrogen Fixation During a Diatom Bloom: Insights and Uncertainties

In our metagenomic analysis, we filtered the *nif* H, *nif* D, *nif* K genes, which code for the nitrogenase enzyme responsible for catalyzing $N_2$ fixation. We could identify sequences related to UCYN-A, which dominated the sequence pool of diazotrophs, particularly in the upper water masses (0 to 5 m) (Fig. 4). UCYN-A, a unicellular cyanobacterial symbiont, has a cosmopolitan distribution and is thought to substantially contribute to global $N_2$ fixation, as documented by (Martínez-Pérez et al., 2016; Tang et al., 2019). This conclusion is based on our metagenomic analysis, in which we set a sequence identity threshold of 95% for both *nif* and photosystem genes. Notably, we only recovered sequences related to UCYN-A

within our *nif* sequence pool, suggesting its predominance among detected diazotrophs. However, metagenomic approaches may underestimate overall diazotroph diversity, and we cannot fully exclude the presence of other, less abundant diazotrophs that may have been missed using this method. While UCYN-A was primarily detected in surface waters, we also observed relatively high *nifK* values at S3_100m, an unusual finding given that UCYN-A is typically constrained to the euphotic zone. Previous studies have predominantly reported UCYN-A in surface waters; for instance Harding et al. (2018) and Shiozaki et al. (2017) detected UCYN-A exclusively in the upper layers of the Arctic Ocean. Additionally, Shiozaki et al. (2020) found UCYN-A2 at depths extending to the 0.1% light level but not below 66 m in the Chukchi Sea. The detection of UCYN-A at 100 m in our study suggests that alternative mechanisms, such as particle association, vertical transport, or local environmental conditions, may facilitate its presence at depth. Interestingly, despite very low *nifH* copy numbers being reported in nearby Baffin Bay by Robicheau et al. (2023), UCYN-A dominated the metagenomic *nifH* community in our study, further underscoring this organisms's presence in Arctic surface coastal areas under certain environmental conditions. This warrants further investigation into the environmental drivers and potential processes enabling its occurrence in Arctic waters.

 Due to the lack of genes such as those encoding Photosystem II and Rubisco, UCYN-A plays a significant role within the host cell and participates in fundamental cellular processes. Consequently, it has evolved to become a closely integrated component of the host cell. Very recent findings demonstrate that UCYN-A imports proteins encoded by the host genome and has been described as an early form of $N_2$ fixing organelle termed a "Nitroplast" (Coale et al. (2024)).

Previous investigations document that they are critical for primary production, supplying up to 85% of the fixed nitrogen to their haptophyte host (Martínez-Pérez et al. (2016)). In addition to its high contribution to primary production, studies have shown that UCYN-A in high latitude waters fix similar amounts of $N_2$ per cell as in the tropical Atlantic Ocean, even in nitrogen- replete waters (Harding et al., 2018; Shiozaki et al., 2020; Martínez-Pérez et al., 2016; Krupke et al., 2015; Mills et al., 2020). However, estimating their contribution to $N_2$ fixation in our study is challenging, particularly since we detected cyanobacteria only at the surface but observe significant $N_2$ fixation rates below 5 m. The diazotrophic community is often underrepresented in metagenomic datasets due to the low abundance of nitrogenase gene copies, implying our data does not present a complete picture. We suspect a more diverse diazotrophic community exists, with UCYN-A being a significant contributor to $N_2$ fixation in Arctic waters. However, the exact proportion of its contribution requires further investigation. The contribution of $N_2$ fixation to carbon fixation (as percent of PP) is relatively low, at the time of our study. We identified genes such as *rbc*L, which encodes Rubisco, a key enzyme in the carbon fixation pathway and *psb*A, a gene encoding Photosystem II, involved in light-driven electron transfer in photosynthesis, in our metagenomic dataset. The gene *rbc*L (for the carbon fixation pathway) and the gene *psb*A (for primary producers) were used to track the community of photosynthetic primary producers in our metagenomic dataset. At station 7, elevated carbon fixation rates are correlated with high diatom (*Bacillariophyta*) abundance and increased chl *a* concentration (Fig. 4), suggesting the onset of a bloom, which is also observable via satellite images (Appendix A1). We hypothesize that meltwater, carrying elevated nutrient and trace metal concentrations, was rapidly transported away from the glacier through the Vaigat Strait by strong winds, leading to increased

productivity, as previously described by Fox and Walker (2022) & Jensen et al. (1999). The elevated diatom abundance and
primary production rates at station 7 coincide with the highest N₂ fixation rates, which could point toward a possible diatom-
diazotroph symbiosis (Foster et al., 2022, 2011; Schvarcz et al., 2022). However, we did not detect a clear diazotrophic signal
directly associated with the diatoms in our metagenomic dataset, which might be due to generally underrepresentation of
diazotrophs in metagenomes due to low abundance or low sequencing coverage. To investigate this further, we
examined the taxonomic composition of *Bacillariophyta* at higher resolution. Among the various abundant diatom
genera, *Rhizosolenia* and *Chaetoceros* have been identified as symbiosis with diazotrophs (Grosse, *et al.*, 2010;
Foster, et al., 2010), representing less than 6% or 15% of *Bacillariophyta*, based on *rbc*L or *psb*A, respectively
(Figure Appendix A4). Although we underestimate diazotrophs to an extent, the presence of certain diatom-
diazotroph symbiosis could help explain the high nitrogen fixation rates in the diatom bloom to a certain degree.
Compilation of *nif* sequences identified from this study as well as homologous from their NCBI top hit were added
in Table S1. However, we cannot tell if the diazotrophs belong to UCYN-A1 or UCYN-A2, or UCYN-A3. Based on
the Pierella Karlusich et al. (2021), they generated clonal *nif*H sequences from Tara Oceans, which the length of *nif*H
sequences is much shorter than the two *nif*H sequences we generated in our study. Also, the available UCYN-A2 or
UCYN-A3 *nif*H sequences from NCBI were shorter than the two *nif*H sequences we generated. Therefore, it would
be not accurate to assign the *nif*H sequences to either group under UCYN-A. Furthermore, not much information is
available regarding the different groups of UCYN-A using marker genes of *nif*D and *nif*K.

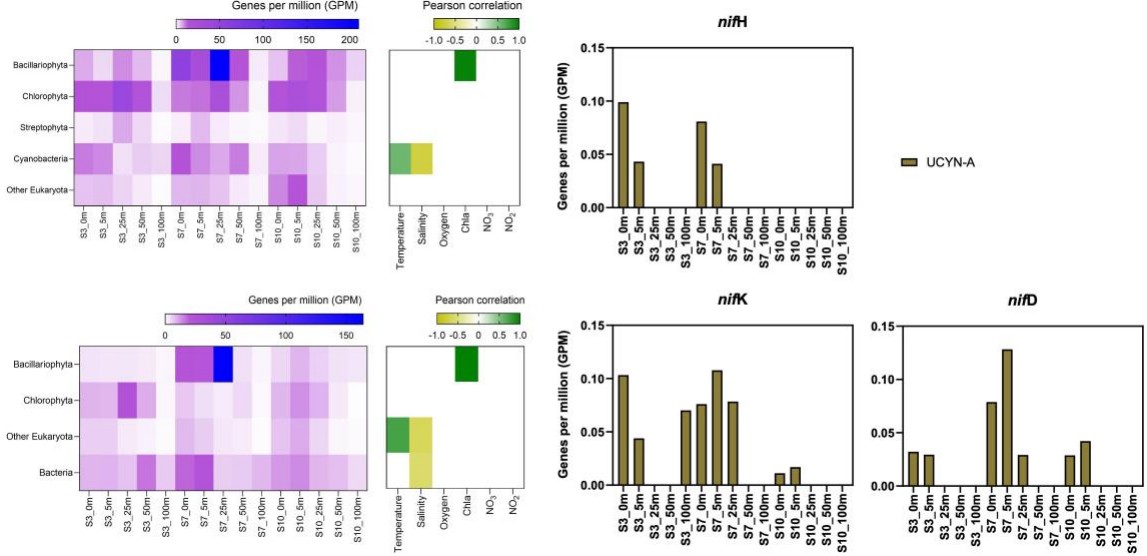


**Figure 4.** Upper left image: *psb*A with correlation plot. Lower left image: *rbc*L with correlation plot. Right image: *nif* H, *nif* D, *nif* K
genes per million reads in the metagenomic datasets. All figures display molecular data from metagenomic dataset for all sampled depth
of station 3,7,10

There is evidence that UCYN-A have a higher Fe demand, with input through meltwater or river runoff potentially being
advantageous to those organisms (Shiozaki et al., 2017, 2018; Cheung et al., 2022). Consequently, UCYN-A might play a
more critical role in the future with increased Fe-rich meltwater runoff. UCYN-A can potentially fuel primary productivity
by supplying nitrogen, especially with increased melting, nutrient inputs, and more light availability due to rising
temperatures as- sociated with climate change. This predicted enhancement of primary productivity may contribute to the
biological drawdown of $CO_2$, acting as a negative feedback mechanism. These projections are based on studies forecasting
increased temperatures, melting, and resulting biogeochemical changes leading to higher primary productivity. However
large uncertainties make pre- dictions very difficult and should be handled with care. Thus, we can only hypothesize that
UCYN-A might be coupled to these dynamics by providing essential nitrogen.
**3.4  $\delta\,^{15}N$ Signatures in particulate organic nitrogen**

Stable isotopic composition, expressed using the $\delta\,^{15}N$ notation, serve as indicators for understanding nitrogen dynamics
because different biogeochemical processes fractionate nitrogen isotopes in distinct ways (Montoya (2008)). However, it
is important to keep in mind that the final isotopic signal is a combination of all processes and an accurate distinction
between processes cannot be made. $N_2$ fixation tends to enrich nitrogenous compounds with lighter isotopes, producing
OM with isotopic values ranging approximately from -2 to +2 ‰(Dähnke and Thamdrup (2013)). Upon complete
remineralization and oxidation, organic matter contributes to a reduction in the average $\delta$-values in the open ocean
(e.g.Montoya et al. (2002);
Emeis et al. (2010)). Whereas processes like denitrification and anammox preferentially remove lighter isotopes, leading
to enrichment in heavier isotopes and delta values up to -25 ‰.

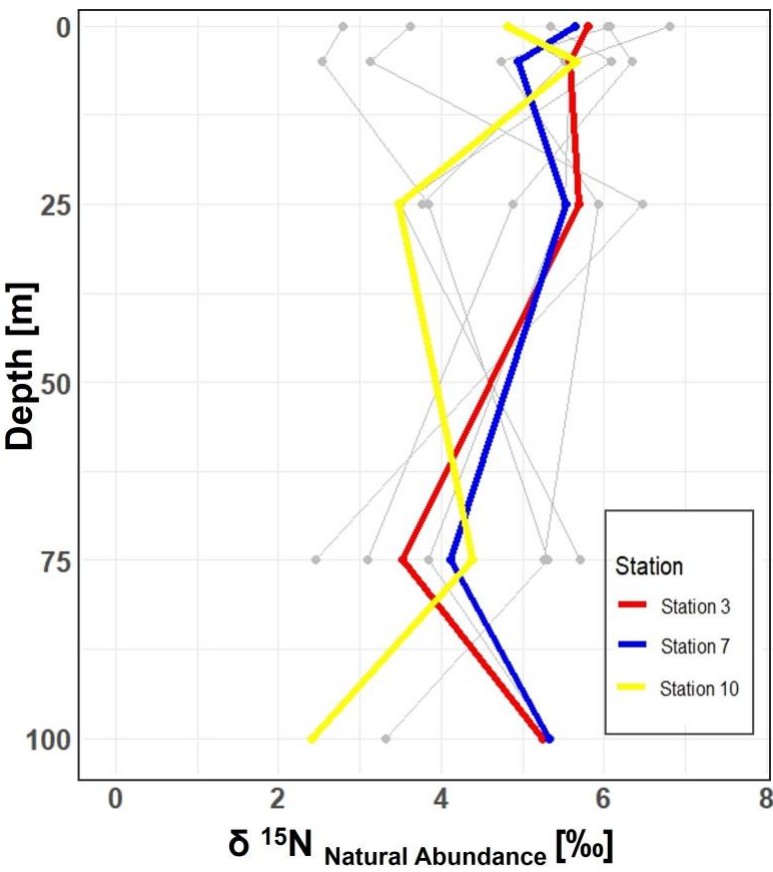

**Figure 5.** Vertical profiles of $\delta$ $^{15}$N natural abundance signatures in PON across 10 stations in the study area. Incubation stations 3, 7, and
10 are highlighted in red, blue, and yellow, respectively. The figure shows variations in $\delta$ $^{15}$N signatures with depth at each station,
providing insight into nitrogen cycling in the study area.
In our study, the $\delta$ $^{15}$N values of PON from all 10 stations, range between 2.45 ‰ and 8.30 ‰ within the 0 to 100 m depth
range. While N$_2$ fixation typically produces OM ranging from -2 ‰ to 0.5 ‰, this signal can be masked by processes such as
remineralization, mixing with nitrate from deeper waters or other biological transformations (Emeis et al. (2010); Sigman
et al. (2009)). The composition of OM in the surface ocean is influenced by the nitrogen substrate and the fractionation
factor during assimilation. When nitrate is depleted in the surface ocean, the isotopic signature of OM produced during
photosynthesis will mirror that of the nitrogen source. Nitrate, the primary form of dissolved nitrogen in the open ocean,
typically exhibits an average stable isotope value of around
5 ‰. No fractionation occurs during photosynthesis because the nitrogen source is entirely taken up in the surface waters
(Sigman et al. (2009)). This matches conditions observed in Qeqertarsuaq, suggesting that subsurface nitrate is a dominant
nitrogen source (Fox and Walker (2022)).
In the eastern Baffin Bay waters, Atlantic water masses serve as an important source of nitrate to surface waters with $\delta^{15}$N
values around 5‰ (Sherwood et al. (2021)). This is consistent with our observed PON values and supports the view that
primary productivity in the region is largely fueled by nitrate input from deeper Atlantic waters, particularly during early
bloom stages (Fox and Walker, 2022; Knies, 2022). The mechanisms through which subsurface nitrate reaches the euphotic
layer are not well understood. However, potential pathways include vertical migration of phytoplankton and physical
mixing. Subsequently, nitrogen undergoes rapid recycling and remineralization processes to meet the system's nitrogen
demands (Jensen et al. (1999)). Taken together, the $\delta^{15}$N signatures observed in this study are best interpreted as indicative
of a system influenced by multiple nitrogen sources and biogeochemical processes, where nitrate input and remineralization
appear to dominate.

**4   Conclusion**

Our study highlights the occurrence of elevated rates of $N_2$ fixation in Arctic coastal waters, particularly prominent at station
7, where they coincide with high chl $a$ values, indicative of heightened productivity. Satellite observations tracing the origin
of a bloom near the Isbræ Glacier, subsequently moving through the Vaigat strait, suggest a recurring phenomenon likely
triggered by increased nutrient-rich meltwater originating from the glacier. This aligns with previous reports by Jensen et
al. (1999) & Fox and Walker (2022), underlining the significance of such events in driving primary productivity in the region.
The contribu- tion of $N_2$ fixation to primary production was low (average 1.57 %) across the stations. Since the demand was
high relative to the new nitrogen provided by $N_2$ fixation, the observed primary production must be sustained by the already
present or adequate amount of subsurface supply of $NO_x$ nutrients in the seawater. This is also visible in the isotopic signature
of the POM (Fox and Walker, 2022; Sherwood et al., 2021). However, the detected $N_2$ fixation rates are likely linked to the
development of the fresh secondary summer bloom, which could be sustained by high nutrient and Fe availability from
melting, potentially leading the system into a nutrient-limited state. The ongoing high demand for nitrogen compounds may
suggest an onset to further sustain the bloom, but it remains speculative whether Fe availability definitively contributes to
this process. The occurrence of such double blooms has increased by 10 % in the Qeqertarsuaq and even 33 % in the Baffin
Bay, with further projected increases moving north from Greenland (Kalaallit Nunaat) waters (Ardyna et al. (2014)). Thus,
nutrient demands are likely to increase, and the role of $N_2$ fixation can become more significant. The diazotrophic community
in this study is dominated by UCYN-A in surface waters and may be linked to diatom abundance in deeper layers. This co-
occurrence of diatoms and $N_2$ fixers in the same location is probably due to the co-limitation of similar nutrients, rather than
a symbiotic relationship. Thus, this highlights the significant presence of diazotrophs despite their limited representation in
datasets. It also highlights the potential for further discoveries, as existing datasets likely underestimate the full extent of
the diazotrophic community (Laso Perez et al., 2024;

Shao et al., 2023; Shiozaki et al., 2017, 2023). The reported $N_2$ fixation rates in the Vaigat strait within the Arctic Ocean are notably higher than those observed in many other oceanic regions, emphasizing that $N_2$ fixation is an active and significant process in these high-latitude waters. When compared to measured rates across various ocean systems using the $^{15}N$ approach, the significance of these findings becomes clear. For instance, $N_2$ fixation rates are sometimes below the detection limit and often relatively low ranging from 0.8 to 4.4 nmol N $L^{-1}$ $d^{-1}$ (Löscher et al., 2020, 2016; Turk et al., 2011). In contrast, higher rates reach up to 20 nmol N $L^{-1}$ $d^{-1}$ (Rees et al. (2009)) and sometime exceptional high rates range from 38 to 610 nmol N $L^{-1}$ $d^{-1}$ (Bonnet et al. (2009)). The Arctic Ocean rates are thus significant in the global context, underscoring the region's role in the global nitrogen cycle and the importance of $N_2$ fixation in supporting primary productivity in these waters.

These findings highlight the urgent need to understand the interplay between seasonal variations, sea-ice dynamics, and hydro- graphic conditions in Qeqertarsuaq. As climate change accelerates the melting of the Greenland Ice Sheet at Jakobshavn Isbræ, shifts in hydrodynamic patterns and hydrographic conditions in Qeqertarsuaq are anticipated. The resulting influx of warmer waters could significantly reshape the bay's hydrography, making it crucial to comprehend the coupling of climate-driven changes and oceanic processes in this vital Arctic region. Our study provides key insights into these dynamics and underscores the importance of continued investigation to predict Qeqertarsuaq's future hydrographic state. By detailing the environmental and hydrographic changes, we contribute valuable knowledge to the broader context of $N_2$ fixation in the Arctic Ocean. Given nitrogen's pivotal role in Arctic ecosystem productivity, it is essential to explore diazotrophs, quantify $N_2$ fixation, and assess their impact on ecosystem services as climate change progresses.

**Appendix A**

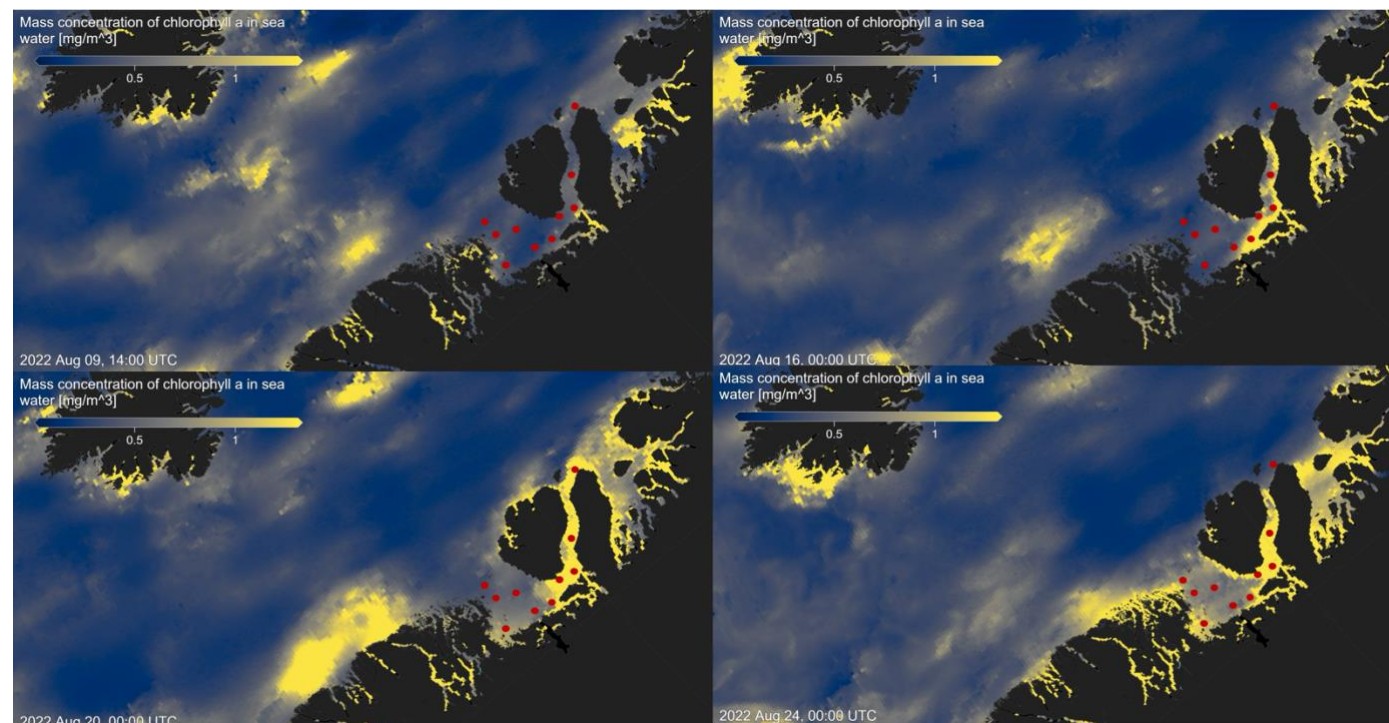

**Figure A1.** Chlorophyll *a* concentration mg m$^{-3}$ at four time points before, during, and after sea water sampling in August 2022 (sampling stations indicated by red dots), obtained from MODIS-Aqua; https://giovanni.gsfc.nasa.gov (Aqua MODIS Global Mapped Chl *a* Data, version R2022.0, DOI:10.5067/AQUA/MODIS/L3M/CHL/2022), 4 km resolution, last access 03 June 2024

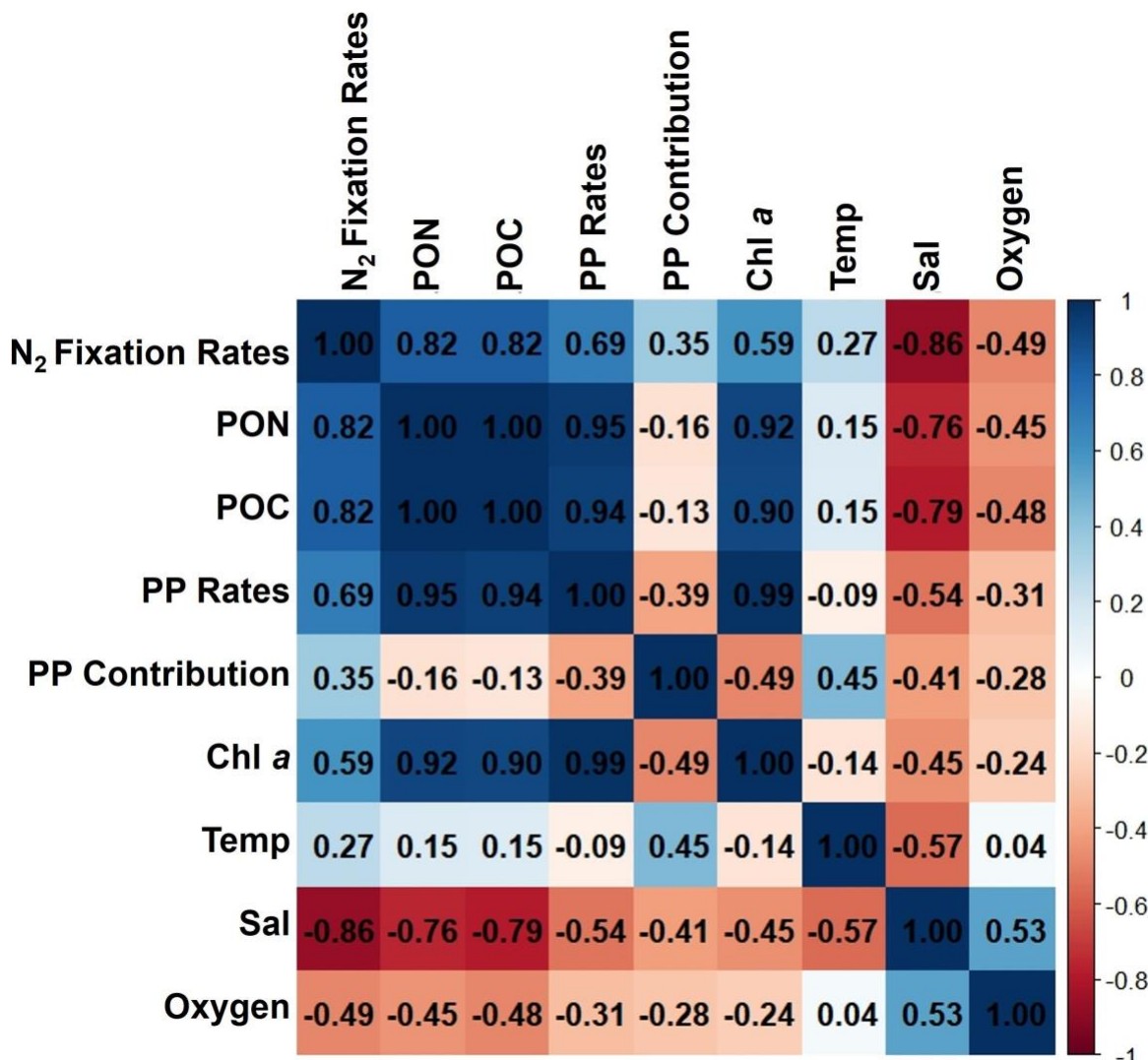

**Figure A2.** Correlation matrix of environmental and biological variables. The plot shows the correlation coefficients between the
following parameters: $N_2$ fixation rates, PON, POC, PP rates, the contribution $N_2$ fixation to PP (PP contribution), Chl $a$, temperature
(Temp), salinity (Sal), and Oxygen. The scale ranges from -1 to 1, where values close to 1 or -1 indicate strong positive or negative
correlations, respectively, and values near 0 indicate weak or no correlation. The color intensity represents the strength and direction of
the correlations, facilitating the identification of relationships among the variables

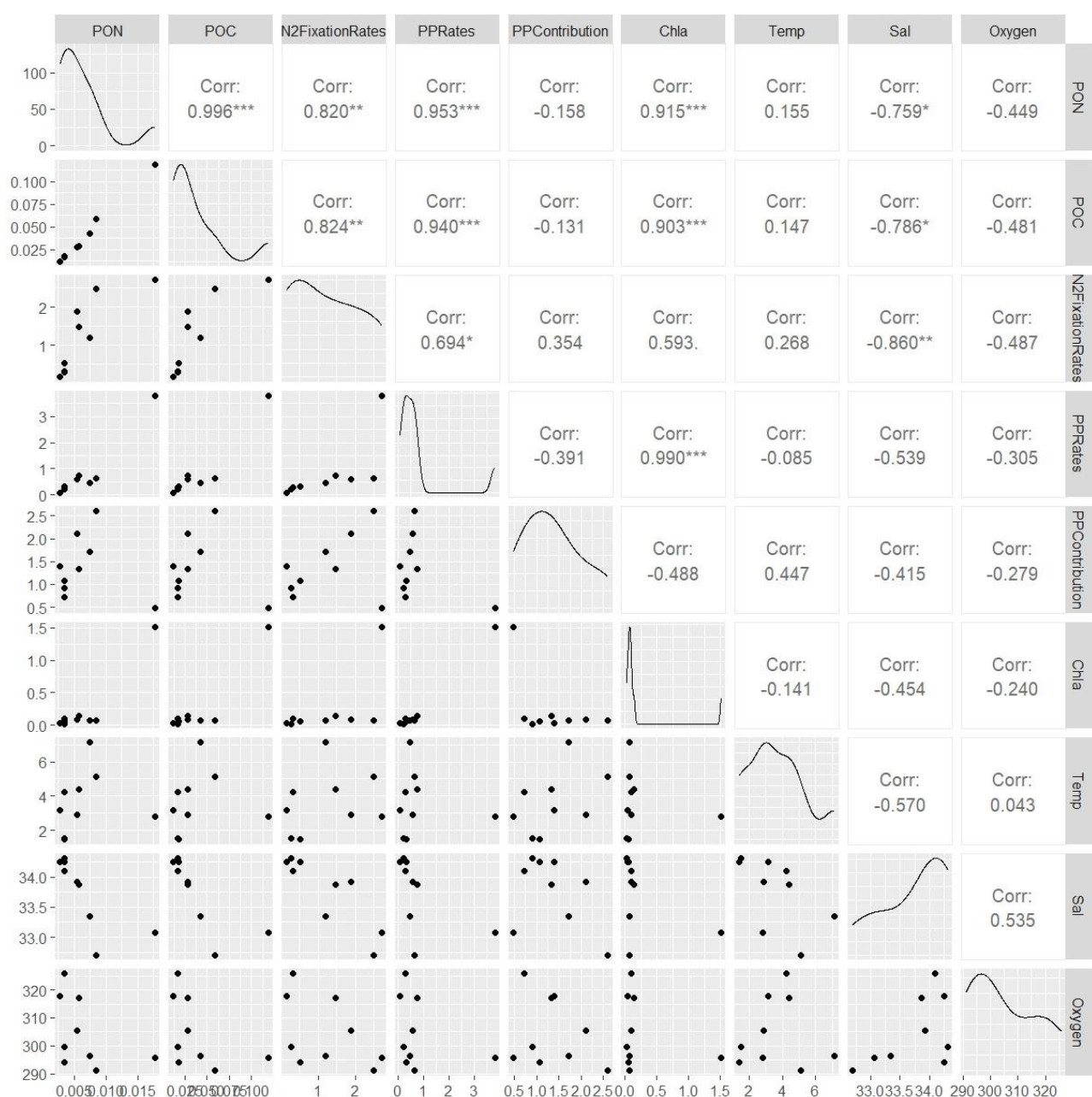

**Figure A3.** This figure displays a ggpairs plot, showing pairwise relationships and correlations between biological and environmental vari- ables. Pearson correlation coefficients displayed in the upper triangular panel, indicating the strength and significance of linear relationships. Statistical significance levels are indicated by stars (*), where * indicates $p < 0.05$, ** indicates $p < 0.01$ and *** indicates $p < 0.001$

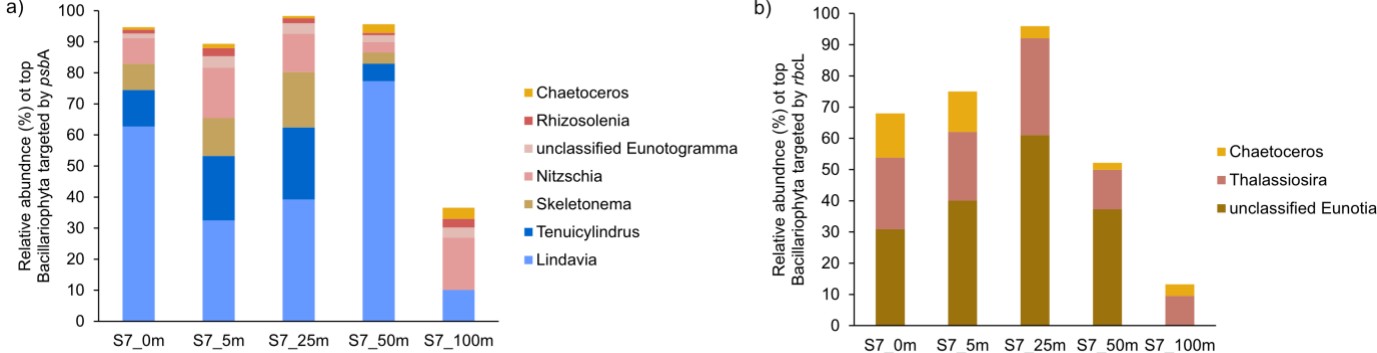

505

**Figure A4 .** Taxonomic composition of Bacillariophyta at Station 7 based on a) psbA and b) rbcL marker genes. The figure shows the relative abundance of Bacillariophyta genera detected in the metagenomic dataset, grouped by gene-specific classifications.

508

| Station | Parameter (X) | Value | SD | $\delta NFR/\delta X$ | Error contribution (SD x $[\delta NFR/\delta X])2$ | % Total error | Summary (nmol N L$^{-1}$ d$^{-1}$) |
|---|---|---|---|---|---|---|---|
| 3 | $\Delta t$ | 1.00 | 0.00 | 0.00 | 0.00 | 0.00 | Mean = 1.13 LOD = 0.73 MQR = 0.12 |
| | $A_{N2}$ | 3.92% | 0.00 | 0.00 | 0.00 | 0.00 | |
| | $A_{PNO}$ | 0.370% | 4.24 x 10$^{-6}$ | 2.63 x 10$^6$ | 2.46 x 10$^2$ | 29.49 | |
| | $A_{PNf}$ | 0.420% | 3.7 x 10$^{-5}$ | 2.36 x 10$^5$ | 3.03 x 10$^2$ | 35.54 | |
| | $[PN]_f$ | 1.69 x 10$^3$ | 1.24 x 10$^2$ | 5.12 x 10$^{-2}$ | 3.21 x 10$^2$ | 34.97 | |
| | | | | | | | |
| 7 | $\Delta t$ | 1.00 | 0.00 | 0.00 | 0.00 | 0.00 | Mean = 1.92 LOD = 1.91 MQR = 0.47 |
| | $A_{N2}$ | 3.92% | 0.00 | 0.00 | 0.00 | 0.00 | |

| | | | | | | | |
|---|---|---|---|---|---|---|---|
| | $A_{PNO}$ | 0.369% | $4.0 \times 10^{-6}$ | $1.57 \times 10^7$ | $2.06 \times 10^3$ | 25.17 | |
| | $A_{PNf}$ | 0.407% | $5.47 \times 10^{-5}$ | $9.25 \times 10^5$ | $2.79 \times 10^3$ | 36.88 | |
| | $[PN]_f$ | $4.62 \times 10^3$ | $8.2 \times 10^2$ | $6.77 \times 10^{-2}$ | $2.87 \times 10^3$ | 37.95 | |
| | | | | | | | |
| 10 | $\Delta t$ | 1.00 | 0.00 | 0.00 | 0.00 | 0.00 | Mean = 0.90 LOD = 0.96 MQR = 0.06 |
| | $A_{N2}$ | 3.92% | 0.00 | 0.00 | 0.00 | 0.00 | |
| | $A_{PNO}$ | 0.371% | $1.89 \times 10^{-6}$ | $-2.01 \times 10^2$ | $1.44 \times 10^{-3}$ | 31.24 | |
| | $A_{PNf}$ | 0.371% | $2.22 \times 10^{-6}$ | $2.01 \times 10^2$ | $2.05 \times 10^{-3}$ | 34.85 | |
| | $[PN]_f$ | $5.91 \times 10^2$ | $1.89 \times 10^2$ | $-1.56 \times 10^{-4}$ | $3.69 \times 10^{-3}$ | 33.91 | |

*Table A1: Sensitivity analysis for $N_2$ fixation rates. The contribution of each source of error to the total uncertainty was determined and calculated after Montoya et al., (1996). Average values and standard deviations (SD) are provided for all parameters at each station. The partial derivative ($\delta NFR/ \delta X$) of the $N_2$ fixation rate measurements is calculated for each parameter and evaluated using the provided average and standard deviation. The total and relative error are given for each parameter. Mean represents the average $N_2$ fixation rate measurement. MQR (minimal quantifiable rate) represents the total uncertainty linked to every measurement and is calculated using standard propagation of error. LOD (limit of detection) represents an alternative detection limit defined as $\Delta APN = 0,00146$.*

*Data availability.* The presented data collected during the cruise will be made accessible on PANGEA.The molecular datasets have been deposited with the accession number: Bioproject PRJNA1133027

*Author contributions.* IS carried out fieldwork and laboratory work at the University of Southern Denmark, and wrote the majority of the manuscript. ELP, AM, and EL conducted fieldwork and laboratory work at the University of Southern Denmark. PX performed metagenomic analysis and created the corresponding graphs. CRL designed the study, provided supervision and guidance throughout the project, and contributed to the writing and revision of the manuscript. All authors contributed to the conception of the study and participated in the writing and revision of the manuscript.

*Competing interests.* The authors declare that they have no known competing financial interests or personal relationships that could have appeared to influence the work reported in this paper. One of the authors, CRL, serves as an Associate Editor for Biogeosciences.

*Acknowledgements.* This work was supported by the Velux Foundation (grant no.29411 to Carolin R. Löscher) and through the DFF grant from the the Independent Research Fund Denmark (grant no. 0217-00089B to Lasse Riemann, Carolin R. Löscher and Stiig Markager). ELP was supported by a postdoctoral contract from Danmarks Frie Forskningsfond (DFF, 1026-00428B) at SDU, and by a Marie Skłodowska- Curie postdoctoral fellowship (HORIZON291 MSCA-2021-PF-01, project number: 101066750) by the European Commission at Princeton University. We sincerely thank the captain and crew of the P540 during the cruise on the Danish military vessel for their invaluable support and cooperation at sea. Our gratitude extends to Isaaffik Arctic Gateway for providing the infrastructure and opportunities that made this project possible. We also acknowledge Zarah Kofoed for her technical support in the laboratory and thank all the Nordcee laboratory technicians for their general assistance.

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
