# Peer review of "Nitrogen Fixation in Arctic Coastal Waters (Qeqertarsuaq, West 2 Greenland): Influence of Glacial Melt on Diazotrophs, Nutrient 3 Availability, and Seasonal Blooms"

_EGUsphere, 2024_

## Author Response (AR1)

We sincerely appreciate the time and effort the reviewers have dedicated to evaluating our manuscript. We are grateful for their constructive feedback and insightful suggestions, which have provided valuable guidance for improving our work. We are pleased to hear that the reviewers find our study intriguing and valuable, particularly in highlighting the relationship between $N_2$ fixation rates and primary production in Arctic coastal waters. We have carefully considered all comments and will revised the manuscript accordingly to enhance its clarity, robustness, and scientific rigor. Below, we provide a point-by-point response to the general comments raised by the reviewers.

**Point by Point Response to Reviewer 1**

1. This study investigated diazotrophy in the coastal area of Baffin Bay in the Arctic Ocean. To my knowledge, three previous studies have already explored diazotrophy in Baffin Bay (Farnelid et al., 2011; Blais et al., 2012; Robicheau et al., 2023, FEMS Microbiology Ecology). Compared to these earlier studies, the novelty of this research is unclear. Additionally, the manuscript appears to be at a very immature stage, and I identified several issues with the study's approach. **Response:** Thank you for your feedback. We acknowledge the previous studies on diazotrophy in Baffin Bay (Farnelid et al., 2011; Blais et al., 2012; Robicheau et al., 2023). However, despite these contributions, our understanding of diazotrophy in the Arctic Ocean remains highly limited, particularly in coastal regions where environmental conditions can vary significantly. Given the ongoing changes in Arctic ecosystems, additional data are crucial for refining our understanding of nitrogen fixation dynamics in these waters. Our study provides novel insights by investigating nitrogen fixation in a previously unexplored region of Baffin Bay using a multi-faceted approach, including direct rate measurements, molecular analyses, and environmental context. This dataset contributes to filling critical knowledge gaps, helping to assess spatial variability and environmental drivers of diazotrophy in Arctic coastal waters. We appreciate the reviewer's concerns and welcome specific suggestions on how to further clarify the novelty and significance of our study.

2. First, I believe the authors have a fundamental misunderstanding about nitrogen fixation in the Arctic. The Arctic Ocean is unique in that it has nutrient-poor water masses despite being located at high latitudes. However, unlike the permanently nutrient-depleted conditions in subtropical oligotrophic oceans, these nutrient-poor water masses in the Arctic are not permanent. Due to mixing driven by low temperatures, nutrients from deeper layers are readily supplied to the surface. Therefore, the authors' claim that "The fixation of dinitrogen… plays a central role in shaping the biological productivity" is incorrect. This misunderstanding

permeates the entire manuscript, leading me to disagree with its overall content and conclusions.

**Response:** We appreciate your insight into the unique characteristics of nutrient dynamics in the Arctic Ocean. We agree that the Arctic is not permanently nutrient-depleted, as nutrient replenishment from deeper waters does occur due to mixing processes. However, as you correctly pointed out, while these nutrient-poor conditions are not constant and can vary seasonally, they can be an important feature in the productive season.

Our study was conducted during the summer, when the surface waters in this region were in a nutrient-depleted state, which is a period often associated with reduced nutrient availability in the upper ocean layers. It is during these periods that nitrogen fixation by diazotrophs, such as UCYN-A, may play a more significant role in contributing new nitrogen to the system, particularly when surface waters are nutrient-limited due to seasonal stratification. The statement regarding nitrogen fixation as a central process in shaping biological productivity is meant to highlight its potential significance during such nutrient-depleted conditions, where the input of new nitrogen can directly influence primary production.

3. I also have concerns regarding the authors' approach to addressing their research objectives. Nitrogen fixation activity was measured at only three out of ten stations, which the authors describe as "strategically chosen." However, is this claim valid? How did the authors determine in advance that these stations represented different water masses before conducting observations?

**Response:** Thank you for your comment. The selection of nitrogen fixation incubation stations was indeed based on an opportunistic sampling approach, constrained by logistical and time limitations. Within these constraints, we aimed to strategically select stations that would represent different environmental conditions and potential nitrogen fixation niches. The chosen stations encompassed key features of the study area, including a glacier-influenced station, an upwelling-associated station, and an open ocean station. While we acknowledge that we could not determine water mass characteristics in advance of the observations, we ensured that the remaining stations were sampled for general oceanographic parameters to provide a broader spatial context for nitrogen fixation. This approach allowed us to obtain a representative dataset despite the practical limitations of fieldwork in this remote region.

4. There is also an issue with the method used to measure nitrogen fixation. Seawater samples were collected from three depths—0, 25, and 50 m. The light environments at these depths are likely to differ significantly. Did the authors assess the light environment in the water column? How did they simulate the in

situ light conditions (L117-118) without investigating the actual light environment?

**Response:** The light environment in the water column was assessed through PAR (Photosynthetically Active Radiation) measurements. For our incubations, we attempted to approximate in situ light conditions by covering samples with foil, as it is typically done, to reduce light exposure, with the 50 m samples incubated in the dark, as no photosynthetically active organisms were expected at this depth.

We acknowledge that while this setup may not perfectly replicate natural conditions, it was designed to be as close to natural as possible given the constraints of shipboard incubations. We will clarify these methodological details in the manuscript and discuss potential implications of these approximations on our nitrogen fixation measurements.

5. Furthermore, the rationale for conducting metagenomic analysis is unclear. This study focuses on nitrogen fixation, so why did the authors not employ an approach specifically targeting the *nif*H gene? The authors state, "Additional molecular approaches would be necessary to enhance our understanding and show a more detailed picture of the diazotrophic community." If they acknowledge this, why did they not include such approaches in their study? Targeting the *nif*H gene would have been more straightforward and relevant than performing metagenomic analysis.

   **Response:** The decision to use metagenomic analysis was driven by both practical and methodological constraints. Due to the naturally low biomass in these Arctic waters, the quantity of extracted DNA was limited. While we initially considered additional targeted approaches such as *nif*H amplicon sequencing, the available DNA was insufficient to conduct these analyses in addition to metagenomics.

   The metagenomic approach allowed us to investigate the broader microbial community, including diazotrophs, within the constraints of our dataset. We acknowledge that a more targeted *nif*H sequencing approach could provide further resolution and highlight this as an important direction for future studies. Still, deriving functional groups including diazotrophs from metagenomes has also an advantage, namely it is the only way of an unbiased assessment. *nif*H amplicon sequencing is by no means quantitative and while it can show us a higher diversity resolution, it can also mislead on quantiative importance of diazotroph subgroups.

6. Furthermore, the structure of the manuscript presents an issue. The authors have combined the Results and Discussion sections, which hinders the clarity

and comprehension of the content. As detailed below, many of the interpretations of the results are unconvincing and lack sufficient support.

**Results:** Thank you for your feedback. We acknowledge that the combined Results and Discussion structure may not align with all reader preferences. However, this format is commonly used in scientific literature and was chosen to present the findings in a structured manner that, in our view, improves readability and coherence. That said, we will carefully consider the reviewer's suggestion and evaluate whether adjustments could enhance the clarity of our interpretations.

Specific Comments :

- L43-50 I find it unclear how the authors have linked environmental changes in the Arctic to nitrogen fixation.
  **Response:** Thank you for your comment. We acknowledge that the link between environmental changes in the Arctic and nitrogen fixation could be made clearer. While increasing primary productivity—driven by reduced sea ice and longer growing seasons—can lead to higher nutrient demand, the role of nitrogen fixation in sustaining Arctic primary production remains uncertain. The factors controlling diazotrophy in these rapidly changing environments are still poorly understood, and the potential impact of ongoing shifts in stratification, nutrient availability, and microbial community composition needs further investigation. In our study, we aimed to address these knowledge gaps by exploring nitrogen fixation patterns in the context of Arctic environmental change. We will revise this section to better emphasize these uncertainties and the need for focused research on this topic.

- L56-58 While it is true that primary production in the Arctic Ocean is often limited by nitrogen availability, the contribution of nitrogen fixation to new production is typically minimal.
  **Response:** Thank you for your comment. We agree that, based on current knowledge, the overall contribution of nitrogen fixation to new production in the Arctic Ocean is considered minimal. However, as indicated in Sipler et al. (2017), the full extent and significance of nitrogen fixation remain uncertain, especially given the limited spatial and temporal coverage of studies in this region. Our

intent is to emphasize the need for further investigations to better quantify its potential role.

- L60 To effectively investigate diazotroph diversity, the authors should focus on targeting the *nif*H gene..
  **Response:** Thank you for your comment. While targeting the *nif*H gene is a well-established approach for studying diazotroph diversity, the primary aim of this study is to understand $N_2$ fixation dynamics and its implications for ecosystem productivity in the rapidly changing Arctic Ocean. Our approach integrates multiple methodologies, including nitrogen fixation rate measurements and metagenomics, to provide a broader perspective beyond just diazotroph diversity.

- L75-77 Should show the detection limits.
  **Response:** Thank you for the suggestion. We will include the detection limits for the nutrient measurements in the Methods section to provide greater clarity and transparency regarding the analytical precision.

- L91-94 It is unclear why filtration was necessary for collecting samples for DIC. Additionally, DIC is not mentioned or analyzed in the results.
  **Response:** Due to technical issues with the equipment, the DIC data were deemed unreliable and were not used in the calculation of primary productivity rates. As a result, this section will be removed from the methodological section to avoid any confusion regarding the role of DIC in our study.

- L99-101 This part should be included at the beginning of "Seawater sampling."
  **Response:** Thank you for the suggestion. We agree that moving this sentence to the beginning of the "Seawater Sampling" section would improve the logical flow of the methodology. We will revise the section accordingly.

- L102-120 As I wrote above, it is not possible to conduct incubation simulating the light environment without first measuring the actual light conditions.
  **Response:** Thank you for your feedback. We understand the concern regarding the light conditions for the incubations. As we conducted our study aboard a military vessel with limited logistical options, we were not able to directly measure light conditions at the sampling depths for every sample. However, we

did assess the light environment using PAR measurements to provide a general understanding of the light conditions in the water column. For the incubation, we aimed to simulate the in situ light environment by covering the samples with foil to reduce light exposure and keeping the 50 m samples in the dark, as no photosynthetically active organisms were expected at this depth. While this may not replicate the exact natural conditions, we believe it is a reasonable approximation given the constraints of the study.

- L134-154 It is unclear why the authors chose to conduct metagenomic analysis.
  **Response:** Thank you for your comment. Due to the limited amount of DNA available from our samples, we faced challenges in conducting multiple molecular analyses. While we initially aimed to perform amplicon sequencing, the DNA yield was insufficient for producing reliable data. Given these constraints, we chose to proceed with metagenomic analysis, as it allowed us to obtain broader insights into the microbial community. This approach also provided a more comprehensive view of the ecosystem's genetic potential. We acknowledge that targeting specific genes like *nif*H could have been a more focused method, but due to the limitations of sample DNA quantity, metagenomics offered a practical solution for our study objectives.

- L157-165 The content written here is not the result. Please show the location of the West Greenland Current and Jakobshavn Isbrae on the map.
  **Response:** Thank you for your comment. We chose to place it in the study area description to provide context for the reader before diving into the results. We will revise the manuscript to include the location of the West Greenland Current and Jakobshavn Isbræ on the map to enhance the clarity and accessibility of this information.

- L168-169 It is unclear how the authors strategically chosed the stations.
  **Response:** Given the logistical constraints, we strategically selected stations to represent a range of environmental conditions and potential nitrogen fixation niches. These included a glacier-influenced station, an upwelling station, and an open ocean station. While we could not determine water mass characteristics prior to the observations, we sampled additional stations for general oceanographic parameters to provide broader spatial context for nitrogen fixation.

- L170-171 What can be said to indicate that there was an influence from the freshwater input?

  **Response:** Thank you for your comment. The surface temperature and salinity measurements indicate the influence of freshwater input, as temperature and salinity profiles in coastal regions can reveal such effects. In our study, the warmer surface temperatures (ranging from 4.5 to 8°C) combined with the relatively low surface salinity (31 to 34) suggest that freshwater may be entering the system, either through river runoff or melting ice, as this can lead to a stratified water column where the warmer, fresher water stays near the surface.

- L172- It is unclear what the author is attempting to convey or the basis on which their statements are made.

  **Response:** Thank you for your comment. Our intent was to present the results of the hydrographic conditions in the study area. The basis for our statements is the data collected during our survey. The observed surface salinity (31–34) is lower than typical open ocean values, suggesting freshwater input, likely from glacial melt or river discharge. Additionally, surface temperatures (4.5–8°C) reflect warming due to solar radiation and possible freshwater influence.

- L180-185 To my eyes, Station 10 also appears to be stratified near the surface.

  **Response:** Thank you for your comment. We agree that Station 10 does show some degree of stratification near the surface, though to a lesser extent than Stations 3 and 7. Our intent was to highlight that, while a weak stratification is present, the overall water column at Station 10 appears more mixed compared to the other stations. We will revise the text to clarify this distinction and ensure that our description accurately reflects the observed profiles.

- L188 There seems to be a lack of objective evidence provided here. What were the characteristics of upwelling?

  **Response:** Thank you for your comment. We acknowledge the importance of providing clear objective evidence for upwelling. Our assessment is based on hydrographic indicators, specifically the elevated salinity, and lower temperatures at the surface at Station 10 compared to other stations. These features are consistent with upwelling, as documented in previous studies (Hansen et al., 2012; Krawczyk et al., 2022). Additionally, the seafloor topography in this area likely facilitates upwelling by disrupting typical circulation, which has been described in the literature.

- L203 I think this title is incorrect.

  **Response**: We disagree, the title reflects our hypothesis that elevated nitrogen fixation rates could contribute to nutrient dynamics and bloom development, which we aim to explore in this study.

- L221 "relatively high nitrogen fixation rates observed may play a role in bloom dynamics" I do not agree with this description.

  **Response:** We appreciate the comment and understand the concern regarding our statement. We acknowledge that the role of nitrogen fixation in bloom dynamics remains uncertain; however, our hypothesis is based on observed nutrient limitations and supported by previous studies (e.g., Sohm et al., 2011; Reeder et al., 2021). While nitrogen fixation did not sustain primary production during our sampling period, it could become more relevant as nitrogen depletion progresses, potentially prolonging bloom productivity. We will clarify that this is a speculative yet plausible mechanism, emphasizing that further research is needed to confirm its role in Arctic bloom dynamics.

- L222-229 I don't understand what the authors are saying.

  **Response:** Thank you for your comment. This section discusses how the observed N:P ratio suggests nitrogen limitation, which aligns with previous studies in the region (Jensen et al., 1999; Tremblay & Gagnon, 2009). Under such conditions, nitrogen-fixing organisms could occupy a niche (Sohm et al., 2011). While our results suggest that nitrogen fixation did not primarily sustain primary production during our sampling period, we hypothesize that it may contribute to bloom dynamics by providing nitrogen when availability declines. Additionally, satellite data suggest that a secondary bloom occurred in early August following the annual spring bloom (Ardyna et al., 2014). This double bloom scenario could be linked to glacier meltwater inputs, which introduce nutrients and iron (Fe), a key limiting factor for diazotrophs (Bhatia et al., 2013; Hawkings et al., 2014, 2015; Arrigo et al., 2017; Hopwood et al., 2016). Therefore, we propose that meltwater from the Isbræ glacier may have supported the observed bloom and provided a suitable niche for diazotrophs.

- L235-244 I don't understand what the authors are discussing.

  **Response:** Thank you for your comment. We acknowledge that this section may require clarification. Our intent was to discuss how nitrogen fixation could play a supplementary role during bloom development, particularly when nitrogen becomes limiting. While previous studies (e.g., Laso Perez et al., 2024) have not detected *nif*H gene copies in Arctic blooms, this is consistent with the generally

low abundance of diazotrophs in marine microbial communities (Turk-Kubo et al., 2015; Farnelid et al., 2019). However, our findings indicate the presence of diazotrophs in Qeqertarsuaq, suggesting that they may play a more significant role in nitrogen cycling during bloom periods than previously thought. We will revise this section for better clarity.

- L253-255 This is a story about the subtropics. I don't think the Arctic is the same.
**Response:** Thank you for your comment. We acknowledge that the correlation between chl a concentrations and $N_2$ fixation rates has been primarily studied in subtropical regions. However, the relationship between phytoplankton biomass and nitrogen fixation in the Arctic remains less well understood. Based on our data, we observed the highest $N_2$ fixation rates at the station with the highest chl a concentration, suggesting a possible link between local phytoplankton abundance and nitrogen fixation activity in this region. While we do not claim that this correlation is universally applicable to the Arctic, our findings support the hypothesis that $N_2$ fixation may contribute to primary production in high-latitude environments under specific conditions, such as meltwater-induced blooms.

- L260-300 This chapter relies too heavily on arbitrary interpretations, making it difficult to follow the argument. While it is indeed interesting that the *nif* gene of UCYN-A was detected in the metagenome analysis, it is unclear why psbA and rbcL are discussed in relation to nitrogen fixation. These genes are not directly associated with nitrogen fixation processes. Furthermore, the statement at L264-265 is evidently incorrect and requires revision or clarification.
**Response:** We understand that psbA and rbcL are not directly related to nitrogen fixation. Our intention in discussing these genes was not to imply a direct mechanistic link but rather to track primary producer communities in the metagenomic dataset. Specifically, we used *rbc*L to examine carbon fixation pathways and *psb*A to track photosynthetic primary producers, providing additional context for the observed ecosystem dynamics. These genes serve as indicators of phytoplankton abundance and activity, which could be relevant when considering the broader ecological setting in which nitrogen fixation occurs. To improve clarity, we will refine the discussion, ensuring a clear distinction between hypotheses and established findings while keeping a strong focus on our presented data and available literature.
We will revise the statement at L264-265 for accuracy and ensure our discussion remains clear.

- L301-330 The authors state that there was no δ15N signal of nitrogen fixation in the particulate organic carbon (POC). This indicates that the contribution of nitrogen fixation to new production was minimal. Consequently, this finding contradicts the authors' claim of a significant contribution of nitrogen fixation to primary production.

  **Response:** We believe, this is supposed to adress the absence of a clear δ15N signal of nitrogen fixation in PON (instead of POC). Thank you for pointing this out. We agree that the lack of a clear δ15N signal in the PON suggests that nitrogen fixation was not the dominant process in new production at the time of sampling. However, as highlighted in the manuscript, the mixed δ15N signal indicates that multiple processes, including nitrogen fixation, are likely occurring simultaneously. While nitrogen fixation may not dominate the PON signal, the measured nitrogen fixation rates confirm that it is still contributing to primary production in the region. Given the ongoing primary production and increasing nutrient demand, we hypothesize that nitrogen fixation may play a more significant role in future primary productivity in the Arctic.

- L332-367 The conclusion is overly detailed. The key findings of this study can be summarized in 2-3 sentences, focusing on the main highlights. Additionally, the implications of the results and potential future research directions should be stated concisely.

  **Response:** Thank you for your comment. We believe that the information provided is useful to contextualise the insights of our study.

**Point by Point Response to the Reviewer 2**

Specific Comments:

Figure 2: The authors should strongly consider also plotting PAR and Fluorescence that would have been obtained from the CTD sensors in Fig 2. Fluorescence would be especially useful to plot given the high chlorophyll value obtained for station 7 around 25 m depth.

**Response:** We appreciate the reviewer's suggestion to include both PAR and fluorescence data in Figure 2. While fluorescence data were recorded, the sensor was not properly calibrated, leading to unreliable measurements. As a result, we have chosen not to include these data in our analysis. However, we agree that PAR data would provide valuable additional context and has been incorporated into Figure 2 alongside the other CTD parameters.

Lines 209-210: The authors should also draw comparisons to nitrogen fixation rates given in Sipler et al. (2017), especially since some of their nitrogen fixation rates for coastal Alaskan Arctic are higher than those given by the authors. I.e., the authors should also report their rates in relation to overall maximum Arctic rates observed thus far in the literature.

**Response:** We have incorporated a comparison with the nitrogen fixation rates reported by Sipler et al. (2017), highlighting how our rates compare to the maximum observed rates in Arctic coastal environments. This provides a broader context for our findings within the existing literature.

Line 247: In figure 3 (left panel) there are PON values plotted near PON = 0 uM and POC = 0.5 uM, meanwhile, PON range is reported in the main text as 0.5 - 4.0 uM. This discrepancy should be addressed.

**Response:** Thank you for pointing out this discrepancy. The PON range of 0.5–4.0 µM reported in the text specifically refers to the incubation stations, whereas the full dataset, as shown in Figure 3, has a broader range of 0–4.0 µM. To maintain consistency with the figure, we have revised the reported range in the main text accordingly.

Line 251: The authors should provide further information and/or citations for inferring the DCM between 15-30m given that they state that they identified it between such depths. Perhaps this was inferred from CTD fluorescence data?

**Response:** The DCM depth range (15–30 m) was identified based on Chl a concentration profiles rather than fluorescence data. We have clarified this in the text and provided appropriate references to support the methodology.

Lines 254-255: The discussion point on nitrogen fixation rates correlating to chl a concentration is interesting, however, the authors should also for greater context point out that their nitrogen fixation rates versus chl a concentration correlation was not significant overall (see Fig A2 & A3). As the authors go on to explain, Station 7 at 25m was a more unique case.

**Response:** We acknowledge that while there is an interesting trend between nitrogen fixation rates and Chl a concentration, the overall correlation was not

statistically significant, as shown in Figures A2 and A3. We have clarified this in the text to ensure a more balanced interpretation of the results. Additionally, we emphasize that the high nitrogen fixation rate observed at Station 7 (25 m) represents a unique case, which we discuss in more detail.

Lines 256-257: It is difficult to orient the images in Figure A1 to the maps in Fig 1. It would be useful if the sampling station(s) were labelled in Figure A1 and/or if an inset diagram was provided similar to what was given by the first two panels in Figure 1.
**Response:** Thank you for the suggestion. We agree that labeling the sampling stations in Figure A1 would improve clarity and make it easier to relate these images to the maps in Figure 1. We have updated Figure A1 to include station labels, ensuring better orientation for the reader.

Line 262: Although I agree that the UCYN-A was detected mainly at 0-5m depths, the authors should also provide some hypotheses or context as to why relatively high UCYN-A nifK values were detected at S3_100m. Have UCYN-A been detected below the euphotic zone in the Arctic previously?
**Resonse:** Thank you for this valuable comment. While UCYN-A was primarily detected at 0–5 m, we acknowledge the relatively high nifK values observed at S3_100m. We have reviewed the literature and found that previous studies have primarily detected UCYN-A in surface waters. Harding et al. (2018) reported UCYN-A lineages mainly in surface waters, while Shiozaki et al. (2017) also found UCYN-A only in surface samples in the Arctic. Further, Shiozaki et al. (2020) detected UCYN-A2 from the surface to the 0.1% light depth but did not observe it below 66 m in the Chukchi Sea (western Arctic Ocean). Given that UCYN-A has not been previously detected below the euphotic zone, our findings at S3_100m warrant further investigation into potential mechanisms facilitating its presence at this depth.

Line 261: The claim that UCYN-A dominated the sequence pool of diazotrophs requires further context and data presentation before readers can fully assess the conclusion given. For instance, was the UCYN-A signature > 50% of each of the nif genes? Were there many other nif detected (i.e., what was the proportion of UCYN-A vs. other diazotrophs)?
**Response:** This conclusion is based on our metagenomic analysis, the sequence identiy was set to 95% for nif and also for photosystem genes. We could only recover sequences related to UCYN-A in our nif sequence pool. Metagenome analysis,

however, might underestimate diazotrophs altogether and we can't fully exclude that we missed ,using this approach, other less represented diazotrophs. In order to provide a better understanding of the abundance of UCYN-A, we will add qPCR data for UCYN-A1 and UCYN-A2 to the revised version of the manuscript.

Line 261: In regard to the broader statement 'sequences related to UCYN-A', it is known that there are many ecotypes of UCYN-A which can co-exist in coastal regions (e.g., Robicheau et al. (2023a) showed this in the not-too-distant coastal Northwest Atlantic). Overall, it would be very useful to UCYN-A researchers if the authors were to give some sort of assessment for which ecotype(s) they have observed at their study sites.

**Response:** We appreciate the reviewer's insightful comment regarding the diversity of UCYN-A ecotypes and agree that providing ecotype-specific information would indeed be valuable for the research community. Unfortunately, due to unforeseen circumstances related to the severe illness of a co-author responsible for this part of the analysis, we are currently unable to access and include the qPCR data for UCYN-A1 and UCYN-A2 that was initially planned for the revised manuscript.

Lines 260-265: It is also interesting that UCYN-A dominated the metagenomic nif community given that Robicheau et al. (2023b) showed very low nifH copy numbers in the nearby Baffin Bay, further underscoring this organism's presence in Arctic surface coastal areas under certain environmental conditions.

**Response:** We appreciate this insight and have clarified this contrast in the text to highlight the differences in nifH abundance across different Arctic regions, providing context for the potential environmental factors that might explain these discrepancies. We have also included Robicheau et al. (2023b) as a relevant reference in this discussion.

Lines 273-276: The authors should clarify if any other diazotroph sequences were found. At present the results provide a comprehensive assessment of UCYN-A but give little information about any other diazotrophs that may have been detected in the metagenomes.

**Response:** From our metagenomes, we indeed only recovered nif sequences related to UCYN-A, which possibly might result from an underestimation of low abundant sequences in the metageome dataset. We will clarify this point in the manuscript.

287-290: Regarding the high Bacillariophyta values and diatom-diazotroph symbiosis concepts, although this is an interesting hypothesis, if the blooming diatom had such a large psbA and rbcL signature would it not also logically be expected to have a relatively high nif signature if each individual diatom cell within the bloom was indeed involved in symbiosis with a diazotroph? The authors should clarify if the general principal of diazotrophs being poorly sequenced in metagenomes is enough to completely overshadow a strong diatom-diazotroph bloom. It would also likely be informative if the authors provided some sort of assessment for the identity of the diatom beyond the 'Bacillariophyta' taxonomy name given. For instance, if the genus can be identified, is the group known to contain diatom-diazotroph symbiotic species? Also, the authors should give more information about all the nifH, nifK, and nifD genes identified (not just for UCYN-A) for S7_25m. It would also be relevant to examine the results in Pierella Karlusich et al. (2021) who showed diazotroph metagenomics data for the Arctic Ocean as part of their global analysis.

**Response:** Thank you for this comment. As suggested, deeper taxonomic level (in class and genus) of Bacillariophyta are now given in Table S2 and Figure S1. Among the various abundant diatom genera, Rhizosolenia and Chaetoceros have been identified as symbiosis with diazotrophs (Grosse, *et al*., 2010; Foster, et al., 2010), representing less than 6% or 15% of Bacillariophyta, based on *rbc*L or *psb*A, respectively. Although we underestimate diazotrophs to an extent, the presence of certain diatom-diazotroph symbiosis could help explain the high nitrogen fixation rates in the diatom bloom to a certain degree.

Compilation of *nif* sequences identified from this study as well as homologous from their NCBI top hit were added in Table S1. However, we cannot tell if the diazotrophs belong to UCYN-A1 or UCYN-A2, or UCYN-A3. Based on the Pierella Karlusich et al. (2021), they generated clonal nifH sequences from Tara Oceans, which the length of *nif*H sequences is much shorter than the two *nif*H sequences we generated in our study. Also, the available UCYN-A2 or UCYN-A3 *nif*H sequences from NCBI were shorter than the two *nif*H sequences we generated. Therefore, it would be not accurate to assign the *nif*H sequences to either group under UCYN-A. Furthermore, not much information is available regarding the different groups of UCYN-A using marker genes of *nif*D and *nif*K.

- Grosse, J., Bombar, D., Doan, H. N., Nguyen, L. N., & Voss, M. (2010). The Mekong River plume fuels nitrogen fixation and determines phytoplankton species distribution in the South China Sea during low and high discharge season. *Limnology and Oceanography*, *55*(4), 1668-1680.
- Foster, R. A., Goebel, N. L., & Zehr, J. P. (2010). ISOLATION OF CALOTHRIX RHIZOSOLENIAE (CYANOBACTERIA) STRAIN SC01 FROM CHAETOCEROS (BACILLARIOPHYTA) SPP. DIATOMS OF THE SUBTROPICAL NORTH PACIFIC OCEAN 1. *Journal of Phycology*, *46*(5), 1028-1037.

Section 3.3. title: Without further information regarding the entire diazotroph community derived from metagenomics it is hard to infer that UCYN-A might contribute to the nitrogen fixation during the diatom bloom observed. The authors themselves also suggest that maybe a diatom-diazotroph symbiosis might be at play, therefore it seems that this title is a bit misleading with respect to the limitations of the results.

**Response:** We understand the concern regarding the current title. A more precise title could emphasize the uncertainty and limitations of the findings, while still acknowledging the observed patterns. "Potential Contribution of UCYN-A to Nitrogen Fixation During a Diatom Bloom: Insights and Uncertainties"

**Technical Corrections**

Line 35: The text "in those waters" is too imprecise. **-> changed to "in arctic waters"**

Line 139: Change "x" letter to a multiplication symbol "×". **-> changed as suggested**

Line 151: Change ", nifH" to "and nifH". **-> changed as suggested**

Line 151-153: Authors should include references for NCBI accession codes cited. **-> changed as suggested**

Line 231: Typographical error; change "3indicates" to "indicates". **-> changed as suggested**

Line 355: Typographical error; change "?" to a citation or delete question mark. **-> changed as suggested**

**Point by Point Response to the Reviewer 3**

**Major Comments:**

1. **Introduction:**
   **Comment:** The introduction lacks clarity, especially given the numerous recent studies in the Arctic. What are the specific research gaps in this region? The objectives of this study should be explicitly defined in the context of these gaps.
   **Response:** We appreciate this suggestion and have revised the introduction to

better define the research gaps specific to Arctic nitrogen fixation and primary production. We will incorporated a more detailed discussion of recent studies and emphasized the novel aspects of our research. Additionally, we will explicitly stated the study objectives in the context of these gaps to provide a clearer rationale for our work.

2. **Data Analysis:**
   **Comment:** The statistical analysis requires more robustness. Considering the low rates reported, the authors should calculate the minimum quantifiable rates for $N_2$ fixation using standard error propagation methods based on observed variability between replicates, as demonstrated by Gradoville et al. (2017) and the studies afterwards.
   **Response:** We have incorporated the alternative estimate of the limit of detection (LOD) following the method described by Montoya et al. (1996). To calculate the limit of detection (LOD) we set [APNf−APN0] equal to 0.00146 atom %. Based on this method, the smallest detectable difference in enrichment we observed was 0.01448 atom %. Any $N_2$ fixation rates in our study corresponding to differences above this threshold and can be considered reliable and quantifiable.

3. **Nutrient Ratios Discussion:**
   **Comment:** The discussion on nutrient ratios is misleading, as most of the NOx and phosphate data appear to be below detection limits. The N:P ratios of 0/0 should be labeled as "undetermined" rather than zero in the table. Regardless, it is recommended to avoid calculating or discussing N:P ratios under these circumstances.
   **Response:** We appreciate this clarification and will update our tables and discussion accordingly. We will replaced N:P ratio values of 0/0 with "undetermined" and remove any calculations or discussions that rely on undetectable nutrient values. The revised text will then ensures that our interpretation of nutrient ratios is accurate and avoids any misleading conclusions.

4. **Section 3.4 "$\delta^{15}N$ signatures in particulate organic nitrogen (PON) does not provide clear evidence of nitrogen fixation"**
   **Comment:** Since N2 fixation rates have already been measured, this section seems redundant. Consider repurposing the **$\delta^{15}N$** data for other relevant analyses if needed.
   **Response:** We appreciate the reviewer's perspective but disagree that this section is redundant. While direct $N_2$ fixation rates have been measured, the

δ15N signatures provide important complementary evidence that nitrogen cycling processes are more complex than fixation alone. The δ15N data illustrate that multiple nitrogen sources and transformations are occurring, resulting in a mixed isotopic signal.

**Specific Comments:**

Line 205: How were the detection limits defined? Please clarify the method.

→ Based on the detection limit of the MS and threshold of minimal detectable values.

Line 277: "The contribution of N2 fixation to carbon fixation (as percent of PP) is relatively low, but may increase with a further onset of bloom periods." Such sentences throughout the manuscript are pure conjectures and have not substantiated to be convicting. Either remove these or substantiate them.

→ We will consider to substantiat them.

Line 236: Replace *N ratio* with *N:P ratio*.

→ We will replace N ratio to N:P ratio as suggested.

Line 231: "3", typo?

→ The typo will be removed.

Line 231: How does the Redfield or non-Redfield ratio indicate that the particulate organic matter (POM) is freshly derived from an ongoing bloom? Provide a more detailed explanation.

→ The elemental composition of particulate organic matter (POM) can provide insights into its origin and degradation state. Freshly produced POM from an ongoing phytoplankton bloom generally follows or closely resembles the classical Redfield ratio (C:N:P = 106:16:1), as phytoplankton tend to assimilate carbon, nitrogen, and phosphorus in relatively consistent proportions during active growth. However, as POM undergoes degradation and remineralization, microbial activity preferentially depletes nitrogen and phosphorus relative to carbon, leading to an elevated C:N or C:P ratio, deviating from Redfield proportions. A near-Redfield ratio in POM suggests recent primary production, whereas non-Redfield ratios (e.g., higher C:N or C:P) indicate older, more processed organic matter that has undergone selective remineralization. Thus, in our study, assessing the elemental composition of POM allows us to infer whether the organic material is freshly derived from an ongoing bloom or has been subject to substantial degradation (2,3,4)

Line 355: Replace "?"

➔ Will be replaced accordingly.

Figure 3(a): The red line representing the Redfield slope appears to have a higher slope than described. Ensure the figure and text are consistent.

➔ We have revised the graph and checked for accuracy. The slope of the red Redfield line is 6.625.

References:

1. Gradoville, M., Bombar, D., Crump, B., Letelier, R., Zehr, J., & White, A. (2017). Diversity and activity of nitrogen-fixing communities across ocean basins. Limnology and Oceanography, 62, 1895–1909. https://doi.org/10.1002/lno.10542
2. Redfield, A. C. (1934). *On the proportions of organic derivatives in sea water and their relation to the composition of plankton* (Vol. 1). Liverpool: university press of liverpool.
3. Sterner, R. W., & Elser, J. J. (2017). Ecological stoichiometry: the biology of elements from molecules to the biosphere. In *Ecological stoichiometry*. Princeton university press.
4. Geider, R. J., & La Roche, J. (2002). Redfield revisited: variability of C [ratio] N [ratio] P in marine microalgae and its biochemical basis. *European Journal of Phycology*, *37*(1), 1-17.

**We thank the reviewers again for their thoughtful comments, which have significantly improved our manuscript. We hope that our revisions sufficiently address all concerns and look forward to their further feedback if needed.**

---

## Author Response (AR2)

**Dear Editor,**

We would like to sincerely thank you and the reviewers for the time and effort devoted to evaluating our revised manuscript. We greatly appreciate the constructive feedback, which has helped us to further improve our work.

We acknowledge the concerns raised regarding some of our statements not being sufficiently supported by the data, and we have carefully revised the manuscript to address these points in detail. Below, we provide a point-by-point response to the issues raised and outline the changes made in the revised version.

**The statement to the effect that 'N2 fixation ... plays a central role in shaping the biological productivity of the Arctic' should be deleted, because it is not supported by the data presented.**

- o This has been changed and reworded to "The fixation of dinitrogen ($N_2$) gas, a biological process mediated by diazotrophs, provides a source of new nitrogen to marine ecosystems and has been increasingly recognized as a potential contributor to nitrogen supply in the Arctic Ocean. "

**The title of section 3.2 should be replaced by a more neutral, descriptive title. As it stands it includes a claim not supported by data. The statement is valid as a hypothesis, but then it should be presented in the Introduction, and then contrasted with the results obtained. The fact that N2 fixation on average contributed 1.6% to estimated N requirements of primary production is, if anything, evidence against N2 fixation playing a role in bloom dynamics at the time of the study.**

- o The title has been changed to a more neutral version "$N_2$ Fixation Rate Variability and Associated Environmental Conditions".

**The result that N2 fixation on average contributed 1.6% of estimated N requirements of primary production should be included in the Abstract, to make it more informative and avoid misunderstandings.**

- o This has been changed and added.

**The sentences in sections 3.2 and 4 claiming a potential link between N2 fixation and the development of a secondary summer bloom should be accompanied by an acknowledgement or a clarification that to prove this link requires new evidence, so that it remains just a possibility with the data available in the manuscript.**

- Sentences in section 3.2 has been rewritten to "Consequently, it is plausible that Fe and nutrients from the Isbræ glacier create favorable conditions for both bloom development and diazotroph activity in Qeqertarsuaq. However, we emphasize that confirming a causal link between $N_2$ fixation and secondary bloom development requires further evidence, such as time-series data on nutrient concentrations, diazotroph abundance, and bloom dynamics. "

- Sentences in section 4 has been rewritten to "This suggests that $N_2$ fixation may contributes only a certain fraction to export production or that it might have begun to play a role in isotopic fractionation during later stages of the bloom. However, due to the limited temporal resolution and lack of direct measurements of N sources over time, we cannot confirm this dynamic. Additional data – including time-series isotopic profiles and turnover measurements of subsurface nitrate and diazotroph activity – would be needed to establish a causal link between $N_2$ fixation and the observed isotopic patterns in the bloom context. "

**Referee #3 pointed out the need to calculate the minimum quantifiable N2 fixation rates. You responded to this comment in your letter of response but the calculations are not shown in the revised version of the manuscript, which should report the actual detection limits for N2 fixation rates in units of nmolN L-1 d-1. See White et al. 2020 ( Limnol. Oceanogr.: Methods 18, 129–147).**

- MQR (minimum quantifiable rates) have been calculated together with a detailed sensitivity analysis and detection limits. This has been added as a table in the Appendix for all stations as well as a supplementary table with the error contributions for all measured parameters.

**Referee #2 pointed out that the study of Robicheau et al. 2023 should be cited. In your response letter, you indicated that this reference would be included in the revised manuscript, but referee #2 notes this is not the case.**

- This has been included into the Reference list.

---

## Author Response (AR3)

**We thank the editor for the opportunity to revise our manuscript and the reviewers for their valuable feedback. Below, we provide a detailed point-by-point response to the remaining concerns raised by Referee 3, with explanations of how each has been addressed in the revised version.**

N:P Ratio: The table title refers to the N:P ratio, yet no such data are presented
The N:P ratio has been removed from the table title.
As previously mentioned, since nutrient values are reported as below detection limits (BDL), the N:P ratio should not be calculated or discussed. Nevertheless, it remains prominently discussed (e.g., lines 249, 274, 277). This is inappropriate and needs to be removed.
This change has been applied throughout the document. The N:P ratio is no longer used as a basis for the discussion.

Reporting of BDL Values: Throughout Table 1 and elsewhere in the manuscript, values listed as "0" should instead be clearly marked as "BDL" to avoid misinterpretation.
The term **BDL** has been introduced for values previously listed as "0".

Section 3.4 – $\delta^{15}N$ Signatures: The heading "$\delta^{15}N$ Signatures in Particulate Organic Nitrogen Show No Clear Evidence of Nitrogen Fixation" is misleading. As the authors acknowledge in their rebuttal, $\delta^{15}N$ values reflect mixed signals from multiple nitrogen sources and transformations. Since direct $N_2$ fixation rates have already been measured, there is no need to draw further conclusions from $\delta^{15}N$ signatures regarding nitrogen fixation. This section should be reframed accordingly.
The section has been rewritten to base the discussion on $\delta^{15}N$ signatures, focusing on nitrogen sources and overall biogeochemical processes in the study area.

Redfield vs. Non-Redfield Ratios: My earlier comment on how the observed Redfield or non-Redfield ratios indicate freshly derived POM from an ongoing bloom was not addressed. This interpretation requires a more nuanced discussion. Many recent studies show a wide range of C:N:P ratios across natural communities, independent of degradation. Communities dominated by fast-growing organisms often have low N:P ratios, while slower-growing communities exhibit higher ratios. Degradation plays a secondary role at most. The authors should reflect this complexity in their discussion.
The mentioned concerns have been addressed, and additional literature has been reviewed. A more nuanced discussion of OM composition is now included.

Decimal Notation in Supplementary Table: The supplementary tables use commas (,) as decimal separators. Please revise all numerical formatting to adhere to standard international conventions (periods for decimals) throughout the manuscript.

This has been changed accordingly.

---

## Author Response (AR5)

**Author Response Statement**

Dear Handling Associate Editor, Emilio Marañón,

No additional changes have been made to the manuscript since the previous revision. The previously submitted point-by-point response documents outline, in chronological order, all modifications implemented during the peer-review process.

Sincerely,
Isabell Schlangen
on behalf of all co-authors